# A population-based cohort study of longitudinal change of high-density lipoprotein cholesterol impact on gastrointestinal cancer risk

Su Youn Nam [1,2] ✉, Junwoo Jo [3] & Chang-Min Cho[1,2]

High-density Lipoprotein Cholesterol (HDL-C) levels have been associated with cancer. In this observational population-based cohort study using data from the Korean National Health Insurance Service system, we investigate the impact of longitudinal changes in HDL-C levels on gastrointestinal cancer risk. Individuals who underwent health examinations in 2010 and 2014 were followed-up through 2021. Among 3.131 million, 40696 gastric, 35707 colorectal, 21309 liver, 11532 pancreatic, 4225 gallbladder, and 7051 biliary cancers are newly detected. The persistent low HDL-C group increases the risk of gastric, liver, and biliary cancer comparing to persistent normal HDL-C group. HDL-C change from normal to low level increases the risk for gastric, colorectal, liver, pancreatic, gallbladder, and biliary cancers. Effects of HDL-C change on the gastrointestinal cancer risk are also modified by sex and smoking status. HDL-C changes affect the gastric and gallbladder cancer risk in age ≥60 years and the pancreatic and biliary cancer risk in age <60 years. Here, we show persistently low HDL-C and normal-to-low HDL-C change increase gastrointestinal cancer risk with discrepancies by sex, smoking status, and age.

High-density lipoprotein cholesterol (HDL-C) is inversely associated with cardiovascular diseases or mortality[1,2] and non-HDL lipid such as triglyceride and low density lipoprotein cholesterol (LDL-C) is positively associated with them[1]. A protective role of HDL-C in cancer has also been suggested, with epidemiological studies reporting an inverse association between HDL-C and subpopulations of breast[3], endometrial[4], lung, colorectal, and liver cancers[5,6]. However, some studies have found no significant association between HDL-C and several cancers such as breast and colorectal cancers[5,7,8]. Recent cohort studies have suggested that low HDL-C levels increased the risk of several individual cancers[9,10] and a meta-analysis has shown a significant inverse association between HDL-C level and overall cancer[11]. The favorable impact of HDL-C on cardiovascular disease and cancer risk appears to be related to anti-atherosclerotic, anti-thrombotic, anti-inflammatory, anti-apoptotic, anti-oxidative, and immune-modulating effects[12].

As mentioned above, baseline HDL-C levels are inversely associated with the risk of several cancers in cohort studies but have no significant association with some cancers. The normal HDL-C at the baseline is often maintained, but the HDL-C may decrease significantly in some persons. Low HDL-C will also be maintained in many cases, but may be improved to normal HDL-C in some people. The effects of these HDL-C changes on cancer development have not been reported. Here, we show the impact of longitudinal changes in HDL-C level on gastrointestinal cancer risk using a large database from the Korean National Health Insurance Service System (NHISS). Several cancer

[1]Department of Internal Medicine, School of Medicine, Kyungpook National University, Daegu, South Korea. [2]Division of Gastroenterology, Kyungpook National University Chilgok Hospital, Daegu, South Korea. [3]Department of Statistics, Kyungpook National University, Daegu, South Korea. ✉e-mail: nam20131114@gmail.com

**Fig. 1 | Study overview. a** Study flow. **b** Three different models according to exposure time. **c** Scheme of high-density lipoprotein cholesterol change category. HDL-C, high density lipoprotein cholesterol. NGHE National General Health Examination. According to the Adult Treatment Panel III (ATP III), normal HDL-C levels were defined as HDL-C ≥ 40 mg/dL in men and ≥50 mg/dL in women. Low HDL-C levels were defined as HDL-C levels of <40 mg/dL in men and <50 mg/dL in women. ΔHDL-C = [(HDL-C at follow-up) - (HDL-C at baseline)].

studies using the Korean NHISS have been published[13,14]. The definition of normal and low HLD-C was defined according to the Adult Treatment Panel III (ATP III)[15]. Furthermore, we conducted interaction analysis between HDL-C change and important cofactors in cancer risk and performed sub-group analyses to determine their associations by sex, smoking status, and age groups.

## Results

### Baseline demographic and laboratory findings

In 2010, 6.18 million underwent both national heath examination and any cancer screening. Cancers before 2010 were excluded and subjects who did not undergo gastric cancer screening. A total of 4.373 million underwent both national heath examination and gastric cancer screening. After the exclusion of subjects with any cancer diagnosed within 1year and those who died within 1year, 4.323 million persons were enrolled at baseline. After excluding non-participants for the national general health examination or absence of HDL-C value in 2014, 3.135 million individuals were eligible. After further excluding unknown sex type, 3,130,795 persons (1,387,648 men [44%] and mean age of 54 years) are eligible [model I] and followed up until 2021 (Fig. 1a). The baseline characteristics of the included and excluded individuals was provided in Supplementary Table 1. Low and normal HDL-C levels were defined according to the ATP III[15]. Baseline characteristics in the final eligible individuals by HDL-C change group were provided in Table 1. The total person year was 34,907,641 and 112,467 patients had newly detected gastrointestinal cancers up to 12 years. Among 3.131 million persons, 40696 gastric, 35,707 colorectal, 21,309 liver, 11,532 pancreatic, 4225 gallbladder, and 7051 biliary cancers were newly detected. Normal HDL-C level at the baseline was maintained in 84% (n = 2,026,087) at follow-up, and HDL-C decreased to low levels in 16% (385,774). Furthermore, low HDL-C level at baseline persisted in 55% (392,532) at follow-up, but HDL-C level increased to the normal range of HDL-C in 45% (326,402).

### Effect of baseline HDL-C on gastrointestinal cancer risk

In the adjusted analysis, normal HDL-C at baseline was associated with a reduced risk of gastric and pancreatic cancer comparing to low HDL-C in overall population men, and women. Gallbladder-biliary cancer risk had no association with baseline HDL-C. A normal HDL-C at baseline reduced the risk of men's liver cancer but had no association with women's liver cancer risk. Both low (HDL-C < 30 mg; adjusted HR [aHR] = 1.69; 95% CI = 1.55–1.84) and high HDL-C (HDL-C ≥ 60 mg; aHR = 1.14; 95% CI = 1.09–1.18) was associated with an increased risk of liver cancer (Supplementary Table 2).

### Impact of HDL-C change on gastrointestinal cancer risk

In the adjusted analysis for potential variables except triglyceride and LDL-C, the risk of gastric, colorectal, and liver cancers was higher in the persistently low HDL-C and normal-to-low HDL-C groups compared to the persistently normal HDL-C group and the risk of pancreatic and gallbladder cancer was higher in normal-to-low HDL-C groups compared to the persistently normal HDL-C group. In the adjusted analysis for all potential variables including triglyceride and LDL-C, these patterns persisted in most GI cancers (Table 2 and Fig. 2a). The persistently low HDL-C and normal-to-low HDL-C groups had a higher risk of gastric and biliary cancers compared to the persistently normal HDL-C group. The persistently low (aHR = 1.17; 95% CI = 1.12–1.23), normal-to-low (aHR = 1.26; 95% CI = 1.21–1.32), and low-to-normal (aHR = 1.07; 95% CI = 1.02–1.12) HDL-C groups had a higher risk of liver cancer compared to the persistently normal HDL-C group. Normal-to-low HDL-C groups had a higher risk of colorectal (aHR = 1.08; 95% CI = 1.05–1.12), pancreatic (aHR = 1.11; 95% CI = 1.05–1.17), and gallbladder (aHR = 1.12; 95% CI = 1.02–1.23) cancer compared to the persistently normal HDL-C group.

### Sensitivity analysis

Our main results were robust to a range of sensitivity analyses for the impact of the exposure period in most cancers (Table 2,

**Table 1 | Baseline Characteristics by HDL-C change**

| | HDL-C change | | | |
|---|---|---|---|---|
| | Low → Low (*n* = 392,532) | Low → Normal (*n* = 326,402) | Normal → Low (*n* = 385,774) | Normal → Normal (*n* = 2,026087) |
| **Person-years** | 4,380,481 | 3,650,000 | 4,284,187 | 22,592,972 |
| Men, no (%) | 95,039 (24.2) | 109,123 (33.4) | 142,190 (36.9) | 1,041,296 (51.4) |
| Age, year, median (IQR) | 56 (49–66) | 55 (48–64) | 56 (48–64) | 52 (46–62) |
| Economic status, median (IQR)[a] | 13 (8–17) | 13 (8–17) | 13 (7–17) | 14 (8–17) |
| BMI, kg/m², median (IQR) | 24.6 (22.8–26.6) | 24.2 (22.4–26.3) | 24.3 (22.4–26.3) | 23.5 (21.6–25.5) |
| HDL-C, mg/dL, median (IQR) | 40 (36–45) | 41 (37–46) | 53 (49–59) | 58 (51–67) |
| Hypertension, no (%) | 152,411 (38.8) | 114,411 (35.1) | 139,631 (36.2) | 600,649 (29.6) |
| Heart disease, no (%) | 20,113 (5.1) | 14,481 (4.4) | 17,366 (4.5) | 67,584 (3.3) |
| Cerebrovascular disease, no (%) | 10,547 (2.7) | 7816 (2.4) | 9347 (2.4) | 36,572 (1.8) |
| Diabetes mellitus, no (%) | 62,299 (15.9) | 40,489 (12.4) | 54,560 (14.1) | 187,565 (9.3) |
| Lipid lowering drug, no (%) | 16,921 (4.3) | 14,047 (4.3) | 15,467 (4) | 69,969 (3.5) |
| Drinking status, no (%) | | | | |
| None | 309,179 (79) | 233,127 (71.6) | 263,352 (68.5) | 1,099,562 (54.4) |
| 1/week | 45,230 (11.6) | 45,114 (13.9) | 53,724 (14) | 349,786 (17.3) |
| 2–3/week | 28,419 (7.3) | 35,345 (10.9) | 47,876 (12.5) | 397,776 (19.7) |
| 4–5/week | 5049 (1.3) | 7058 (2.2) | 11,098 (2.9) | 102,383 (5.1) |
| ≥6/week | 3711 (0.9) | 5079 (1.6) | 8476 (2.2) | 70,799 (3.5) |
| Smoking status, no (%) | | | | |
| Never | 315,997 (80.7) | 243,664 (74.8) | 279,714 (72.7) | 1,285,447 (63.6) |
| Past | 31,638 (8.1) | 36,925 (11.3) | 46,495 (12.1) | 366,018 (18.1) |
| Current | 43,907 (11.2) | 45,079 (13.8) | 58,504 (15.2) | 368,866 (18.3) |
| Family history of gastric cancer | 42,307 (10.9) | 35,194 (10.9) | 41,559 (10.9) | 216,647 (10.8) |
| Moderate activity, no (%)[b] | | | | |
| None | 242,908 (62) | 196,054 (60.2) | 231,501 (60.2) | 1,122,844 (55.6) |
| 1–2 day/week | 73,233 (18.7) | 64,583 (19.8) | 75,521 (19.6) | 443,643 (22) |
| 3–5 day/week | 57,157 (14.6) | 49,683 (15.3) | 57,995 (15.1) | 345,432 (17.1) |
| 6–7 day/week | 18,259 (4.7) | 15,350 (4.7) | 19,562 (5.1) | 107,078 (5.3) |
| Liver factors, no (%) | | | | |
| Any liver disease | 6317 (1.7) | 5493 (1.7) | 6284 (1.7) | 31,118 (1.6) |
| Chronic hepatitis B | 2217 (0.6) | 1861 (0.6) | 2301 (0.6) | 12,113 (0.6) |
| Chronic hepatitis C | 775 (0.2) | 585 (0.2) | 633 (0.2) | 2551 (0.1) |
| Liver cirrhosis | 424 (0.1) | 330 (0.1) | 417 (0.1) | 2041 (0.1) |

*BMI* body mass index, *HDL-C* high density lipoprotein, *IQR* interquartile range.
[a]Economic status is twentile (1–20). 1 is the lowest and 20 is the highest income.
[b]Moderate physical activity refers to "walking or exercising and feeling mild dyspnea for more than 30 min per day." Low HDL-C refers to <40 mg/dL in men and <50 mg/dL in women. Normal HDL-C means ≥40 mg/dL in men and ≥50 mg/dL in women. Missing rate: 1.74% in economic status.

Supplementary Tables 3, 4). The estimated effect sizes and CIs were in model II (Supplementary Table 3) similar to those in the model I, and the two graphs nearly overlapped in gastric, colorectal, liver, pancreatic, and gallbladder cancer risks (Fig. 2a, b), wherein HDL-C change had no association with biliary cancer risk in model II. In adjusted analysis after further exclusion of cancer up to 2nd measurement of HDL-C [model III], the effect of HDL-C change on gastric, colorectal, and liver cancer risks was similar with that in model I, whereas HDL-C change had no significant effect on the risk of pancreatic, gallbladder, and biliary cancer (Supplementary Table 4 and Fig. 2a, c).

**Subgroup analysis by sex**
Interaction analysis showed a significant interaction between sex and HDL change in the risk of gastrointestinal cancers (Supplementary Table 5). Therefore, we performed subgroup analysis by sex. In adjusted analysis, the impact of HDL-C change on individual cancer risk showed a sex discrepancy in most gastrointestinal cancers (Table 3). The persistently low HDL-C (aHR = 1.11; 95% CI = 1.06–1.16), low-to-normal (aHR = 1.07; 95% CI = 1.02–1.13), and normal-to-low

HDL-C groups (aHR = 1.07; 95% CI = 1.02–1.12) showed an increased gastric cancer risk in women, wherein normal-to-low HDL-C groups increased gastric cancer risk in men. Normal-to-low HDL-C group was associated with an increased risk of colorectal and liver cancer in men, wherein persistent low HDL-C and normal-to-low HDL-C group was associated with an increased risk of colorectal and liver cancer in women. Normal-to-low HDL-C and low-to-normal HDL-C groups were associated with an increased risk of pancreatic cancer in men, wherein persistent low HDL-C and normal-to low HDL-C increased risk of pancreatic cancer in women. HDL-C change had no association with the risk of gallbladder and biliary cancer in both sexes.

**Subgroup analysis by smoking status**
Interaction analysis showed a significant interaction between smoking status and HDL change in the risk of several gastrointestinal cancers. Therefore, we performed subgroup analysis by smoking status (Table 4). The persistently low and normal-to-low HDL-C groups showed increased gastric cancer risk in both never smokers and ever-smokers [past/current smokers]. The hazardous impact of persistently

**Table 2 | Gastrointestinal cancer risk by HDL-C change (Model I)**

| HDL-C change | Number of case | Unadjusted HR<br>HR (95% CI) | *p* value | Adjusted HR[b]<br>HR (95% CI) | *p* value | Adjusted HR[c]<br>HR (95% CI) | *p* value |
|---|---|---|---|---|---|---|---|
| **Gastric cancer** | | | | | | | |
| Low → Low | 4764 | 0.92 (0.89, 0.95) | <0.001 | 1.06 (1.03, 1.10) | <0.001 | 1.05 (1.02, 1.09) | <0.001 |
| Low → Normal | 4061 | 0.94 (0.91, 0.97) | <0.001 | 1.04 (1.00, 1.07) | 0.04 | 1.03 (0.99, 1.07) | 0.11 |
| Normal → Low | 5178 | 1.03 (1.00, 1.06) | 0.09 | 1.05 (1.02, 1.09) | <0.001 | 1.05 (1.02, 1.08) | <0.001 |
| Normal➔ Normal | 26,693 | 1 | | 1 | | 1 | |
| **Colorectal cancer** | | | | | | | |
| Low → Low | 4639 | 1.06 (1.03, 1.09) | <0.001 | 1.04 (1.00, 1.07) | 0.03 | 1.02 (0.98, 1.05) | 0.37 |
| Low → Normal | 3580 | 0.98 (0.95, 1.02) | 0.28 | 0.98 (0.95, 1.02) | 0.37 | 0.97 (0.93, 1.00) | 0.08 |
| Normal → Low | 4919 | 1.15 (1.12, 1.19) | <0.001 | 1.09 (1.06, 1.13) | <0.001 | 1.08 (1.05, 1.12) | <0.001 |
| Normal➔ Normal | 22,569 | 1 | | 1 | | 1 | |
| **Liver cancer**[a] | | | | | | | |
| Low → Low | 2591 | 0.99 (0.95, 1.03) | 0.52 | 1.07 (1.02, 1.12) | <0.001 | 1.17 (1.12, 1.23) | <0.001 |
| Low → Normal | 2029 | 0.93 (0.88, 0.97) | <0.001 | 0.99 (0.94, 1.04) | 0.63 | 1.07 (1.02, 1.12) | 0.01 |
| Normal → Low | 3138 | 1.22 (1.18, 1.27) | <0.001 | 1.21 (1.16, 1.27) | <0.001 | 1.26 (1.21, 1.32) | <0.001 |
| Normal➔ Normal | 13,551 | 1 | | 1 | | 1 | |
| **Pancreatic cancer** | | | | | | | |
| Low → Low | 1646 | 1.22 (1.16, 1.29) | <0.001 | 1.05 (0.99, 1.11) | 0.12 | 1.05 (0.99, 1.12) | 0.08 |
| Low → Normal | 1275 | 1.13 (1.07, 1.20) | <0.001 | 1.04 (0.97, 1.10) | 0.25 | 1.04 (0.98, 1.11) | 0.21 |
| Normal → Low | 1660 | 1.26 (1.20, 1.33) | <0.001 | 1.11 (1.05, 1.17) | <0.001 | 1.11 (1.05, 1.17) | <0.001 |
| Normal➔ Normal | 6951 | 1 | | 1 | | 1 | |
| **Gallbladder cancer** | | | | | | | |
| Low → Low | 587 | 1.19 (1.09, 1.30) | <0.001 | 0.98 (0.89, 1.07) | 0.61 | 0.97 (0.88, 1.07) | 0.49 |
| Low → Normal | 465 | 1.13 (1.02, 1.25) | 0.02 | 1 (0.91, 1.11) | 0.95 | 1.00 (0.9, 1.11) | 0.95 |
| Normal → Low | 634 | 1.32 (1.21, 1.44) | <0.001 | 1.13 (1.03, 1.23) | 0.001 | 1.12 (1.02, 1.23) | 0.01 |
| Normal➔ Normal | 2539 | 1 | | 1 | | 1 | |
| **Biliary cancer** | | | | | | | |
| Low → Low | 892 | 1.02 (0.95, 1.10) | 0.54 | 0.94 (0.87, 1.02) | 0.12 | 1.02 (1.00, 1.05) | 0.03 |
| Low → Normal | 703 | 0.97 (0.89, 1.05) | 0.38 | 0.94 (0.87, 1.02) | 0.16 | 0.97 (0.95, 1.00) | 0.02 |
| Normal → Low | 961 | 1.13 (1.05, 1.21) | <0.001 | 1.03 (0.96, 1.11) | 0.44 | 1.11 (1.08, 1.13) | <0.001 |
| Normal➔ Normal | 4495 | 1 | | 1 | | 1 | |

Adjusted HRs and CIs are derived from Cox proportional regression analysis. All statistical tests are two-sided.

Low HDL-C refers to <40 mg/dL in men and <50 mg/dL in women. Normal HDL-C means ≥40 mg/dL in men and ≥50 mg/dL in women.

*CI* confidence interval, *HDL-C* high density lipoprotein cholesterol, *HR* hazard ratio.

[a]Adjusted for age, sex, economic status, body mass index, hypertension, diabetes, cerebrovascular disease, heart disease, smoking status, drinking status, physical activity, use of lipid lowering drug, liver factors (chronic liver disease, chronic hepatitis B, chronic hepatitis C, and liver cirrhosis) and triglyceride.

[b]Adjusted for age, sex, economic status, body mass index, hypertension, diabetes, cerebrovascular disease, heart disease, smoking status, drinking status, physical activity, and use of lipid lowering drug.

[c]Adjusted for age, sex, economic status, body mass index, hypertension, diabetes, cerebrovascular disease, heart disease, smoking status, drinking status, physical activity, *LDL, TG*, and use of lipid lowering drug.

low HDL-C and change from normal to low HDL-C on liver cancer risk was more remarkable in ever-smokers. The persistently low HDL-C group and normal-to low HDL-C group slightly increased the risk of colorectal cancer in never smokers, whereas the normal-to-low HDL-C group increased colorectal cancer risk in ever-smokers.

### Subgroup analysis by combination of sex and smoking
We conducted subgroup analysis using a combination of sex and smoking (Supplementary Table 6). Persistent low, low-to-normal, and normal-to low HDL-C change slightly increased the gastric cancer risk in never-smoking women and persistent low and normal-to low HDL-C change increased the gastric cancer risk in smoking men. These patterns were also observed in pancreatic cancer risk. In men, normal-to-low HDL-C group was associated with an increased risk of CRC in both never-smokers and ever-smokers, whereas in women persistent low and normal-to low HDL-C change increased the CRC risk in never-smokers.

Persistent low, low-to-normal, and normal-to low HDL-C change markedly increased the liver cancer risk in smoking men. Persistent low and normal-to-low HDL-C change increased the liver cancer risk in never-smoking women and persistent low and low-to-normal HDL-C markedly increased the liver cancer risk in never-smoking women. However, the effect size (aHR) was markedly high in smokers than in never smokers in both sex.

### Subgroup analysis by age groups
Interaction analysis showed a significant interaction between age groups and HDL change in the risk of gastrointestinal cancers. In adjusted analysis, the impact of HDL-C change on several gastrointestinal cancers showed an age discrepancy (Table 5). The effect of HDL-C change on the risk of gastric and gallbladder cancer was significant only in age ≥60 years but not significant in age <60 years. The effect of HDL-C change on the risk of liver and colorectal cancer was significant in both age groups. The effect of HDL-C change on the risk

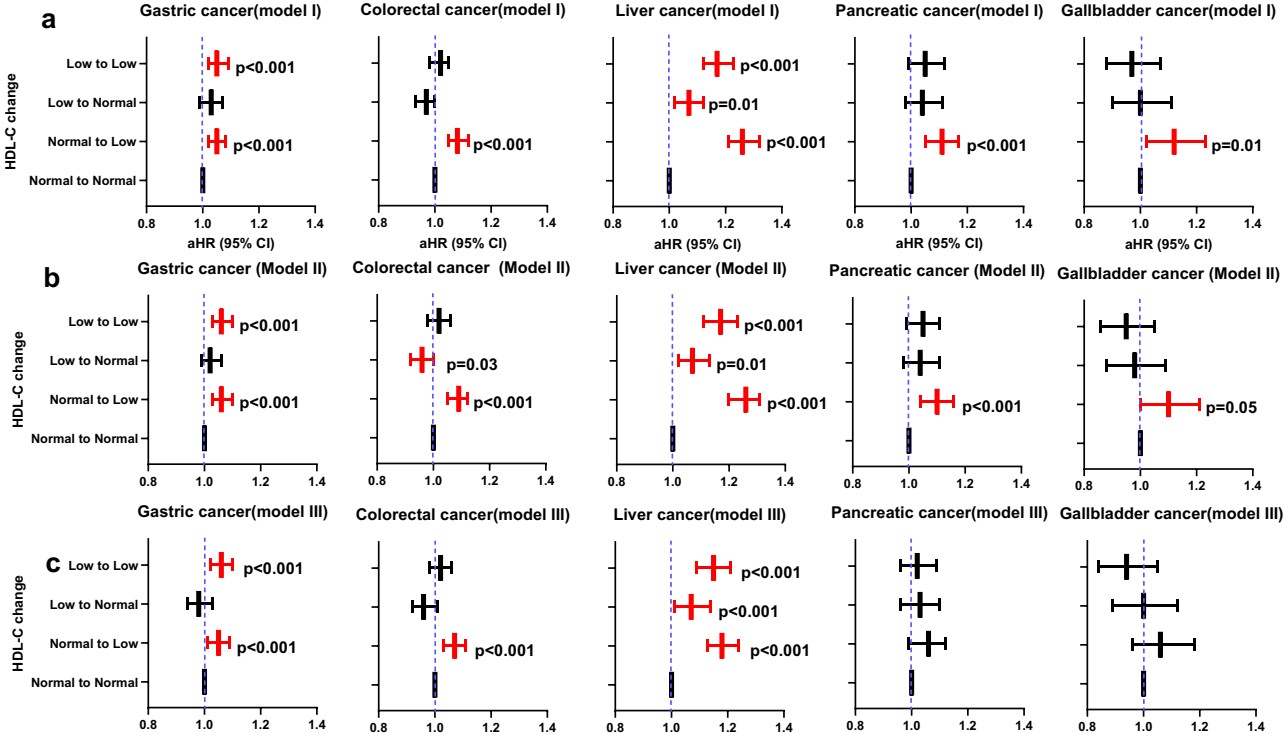

**Fig. 2 | Adjusted hazard ratio by HDL-C change in two models. a** Cancer risk by high density lipoprotein cholesterol (HDL-C) change after excluding any cancer developed within 1 years from first measurement of HDL-C (*model I*). **b** Cancer risk after excluding any cancer developed within 2 years from first measurement of HDL-C (*model II*). **c** Cancer risk after excluding any cancer developed up to second measurement of HDL-C (*model III*). *X*-axis is adjusted hazard ratio (aHR) and *y*-axis is HDL-C change group. Adjusted HRs and CIs are derived from Cox proportional regression analysis. The dashed blue lines illustrate an HR of 1. The reference is a persistently normal (Normal-to-Normal) group. The HR is indicated with the central bar and the error bars show the 95% confidence interval (CI). Red bar is statistical significant HR with CI. Low to Low (n = 392,532), Low to Normal (n = 326,402), Normal to Low (n = 385,774), Normal to Normal (n = 2,026,087) in model I. Low to Low (n = 389,731), Low to Normal (n = 324,102), Normal to Low (n = 382,820), Normal to Normal (n = 2,013,369) in model II. Low to Low (n = 380,934), Low to Normal (n = 698,472), Normal to Low (n = 373,488), Normal to Normal (n = 1,975,069) in model IIII. The exact hazard ratios and 95% confidence intervals are provided in Table 2 and Supplementary Tables 3, 4.

of pancreatic and biliary cancer was significant only in age <60 years, but not significant in age ≥60 years.

### Cancer risk by further categorization of HDL-C change

We assessed gastrointestinal cancer risk according to HDL-C change using ATP III and absolute HDL-C change (Δ HDL-C) to investigate further increase of HDL-C on the cancer risk among baseline normal HDL-C group (Supplementary Table 7 and Fig. 3). Among baseline normal group, an increase >15 mg/dL of HDL-C (aHR = 1.15; 95% CI = 1.08–1.21) as well as normal-to-low HDL-C group (aHR = 1.29; 95% CI = 1.24–1.35) had a higher liver cancer risk compared to the persistently normal group. Among baseline normal group, an increase >15 mg/dL of HDL-C reduced the CRC risk (aHR = 0.94; 95% CI = 1.90–0.99), whereas normal-to-low HDL-C group (aHR = 1.08; 95% CI = 1.05–1.12) had a higher CRC risk compared to the persistently normal group (Fig. 3).

### Effect of absolute HDL-C change on cancer risk

In unadjusted analysis, a decrement of HDL-C at follow-up increases the risk of gastric, colorectal, liver, pancreatic, gallbladder, and biliary cancers comparing to stable HDL-C group. In adjusted analysis, a decrement of HDL-C at follow-up (Δ HDL-C < −10 mg/dL) increases the risk of gastric, colorectal, liver, pancreatic, gallbladder, and biliary cancer (Supplementary Table 8 and Fig. 4). Interestingly, marked increase of HDL-C (Δ HDL-C ≥ 25 mg/dL) also associated with an increased risk of gastric (aHR = 1.07; 95% CI = 1.00–1.14) and liver cancer (aHR = 1.41; 95% CI = 1.3–1.53), whereas increase of HDL-

slightly reduced the risk of colorectal cancer risk (aHR = 0.94; 95% CI = 0.90–0.98).

## Discussion

To the best of our knowledge, this is the first study to investigate the effects of HDL-C level changes on gastrointestinal cancer development. In this large cohort study, persistently low HDL-C had a higher risk for gastric, liver, and biliary cancer compared to persistently normal HDL-C. HDL-C change from normal to low levels increased the risk of gastric, colorectal, liver, pancreatic, gallbladder, and biliary cancers. These results were robust to a range of sensitivity analyses for the impact of the exposure period in most cancers with similar estimated effect sizes. Furthermore, the effect of HDL-C change on the risk of individual gastrointestinal cancer were modified by sex and smoking status. Impact of HDL-C change on the gastric and gallbladder cancer risk was significant only in age ≥60 years, whereas the impact of HDL-C change on the pancreatic and biliary cancer risk was significant only in age <60 years. Effect of HDL-C change on the risk of liver and colorectal cancer was significant in both age groups. Decrement of absolute HDL-C level (ΔHDL-C < −10 mg/dL) increased the risk of all individual digestive cancers, whereas a marked increment of HDL-C had a site-specific discrepancy.

Persistent exposure to low HDL-C and change from normal to low HDL-C increased the risk of gastric cancer and change from normal to low HDL-C increased the risk of colorectal cancer. Further increase of HDL-C level by >15 mg/dL among the baseline normal HDL-C group reduced colorectal cancer risk compared to the persistently normal

**Table 3 | Sub group analysis by sex (adjusted analysis)**

| | HDL-C change | Men | | | | Women | | | |
|---|---|---|---|---|---|---|---|---|---|
| | | Total number | Number of cancer | HR (95% CI)[b] | P value | Total number | Number of cancer | HR (95% CI)[b] | P value |
| Gastric cancer | Low → Low | 95,039 | 1997 | 1.04 (0.99, 1.09) | 0.11 | 297,493 | 2767 | 1.11 (1.06, 1.16) | <0.01 |
| | Low → Normal | 109,123 | 2224 | 1.02 (0.98, 1.07) | 0.38 | 217,279 | 1837 | 1.07 (1.02, 1.13) | 0.01 |
| | Normal → Low | 142,190 | 3105 | 1.05 (1.01, 1.09) | 0.02 | 243,584 | 2073 | 1.07 (1.02, 1.12) | 0.01 |
| | Normal➔ Normal | 1,041,296 | 20,042 | 1 | | 984,791 | 6651 | 1 | |
| Liver cancer[a] | Low → Low | 95,039 | 1141 | 1.06 (0.99, 1.13) | 0.1 | 297,493 | 1450 | 1.08 (1.01, 1.15) | 0.03 |
| | Low → Normal | 109,123 | 1174 | 1.00 (0.94, 1.07) | 0.92 | 217,279 | 855 | 0.97 (0.89, 1.05) | 0.40 |
| | Normal → Low | 142,190 | 1960 | 1.24 (1.18, 1.30) | <0.01 | 243,584 | 1178 | 1.18 (1.09, 1.26) | <0.01 |
| | Normal➔ Normal | 1,041,296 | 10,263 | 1 | | 984,791 | 3288 | 1 | |
| Colorectal cancer | Low → Low | 95,039 | 1483 | 1.04 (0.98, 1.1) | 0.22 | 297,493 | 3156 | 1.06 (1.01, 1.10) | 0.01 |
| | Low → Normal | 109,123 | 1556 | 0.97 (0.92, 1.02) | 0.26 | 217,279 | 2024 | 1.00 (0.95, 1.05) | 0.95 |
| | Normal → Low | 142,190 | 2438 | 1.1 (1.05, 1.15) | <0.01 | 243,584 | 2481 | 1.09 (1.04, 1.14) | <0.01 |
| | Normal➔ Normal | 1,041,296 | 14,770 | 1 | | 984,791 | 7799 | 1 | |
| Pancreatic cancer | Low → Low | 95,039 | 407 | 0.99 (0.89, 1.10) | 0.77 | 297,493 | 1239 | 1.08 (1.00, 1.15) | 0.04 |
| | Low → Normal | 109,123 | 510 | 1.12 (1.02, 1.23) | 0.02 | 217,279 | 765 | 1.00 (0.92, 1.08) | 0.91 |
| | Normal → Low | 142,190 | 691 | 1.10 (1.01, 1.19) | 0.03 | 243,584 | 969 | 1.12 (1.03, 1.20) | <0.01 |
| | Normal➔ Normal | 1,041,296 | 4096 | 1 | | 984,791 | 2855 | 1 | |
| Gallbladder cancer | Low → Low | 95,039 | 118 | 0.92 (0.82, 1.05) | 0.22 | 297,493 | 469 | 1.04 (0.93, 1.16) | 0.51 |
| | Low → Normal | 109,123 | 169 | 0.97 (0.86, 1.09) | 0.55 | 217,279 | 296 | 1.00 (0.88, 1.15) | 0.98 |
| | Normal → Low | 142,190 | 284 | 0.98 (0.89, 1.08) | 0.68 | 243,584 | 350 | 1.04 (0.92, 1.18) | 0.53 |
| | Normal➔ Normal | 1,041,296 | 1476 | 1 | | 984,791 | 1063 | 1 | |
| Biliary cancer | Low → Low | 95,039 | 294 | 0.92 (0.82, 1.05) | 0.22 | 297,493 | 598 | 0.96 (0.87, 1.06) | 0.45 |
| | Low → Normal | 109,123 | 327 | 0.97 (0.86, 1.09) | 0.55 | 217,279 | 376 | 0.93 (0.83, 1.05) | 0.24 |
| | Normal → Low | 142,190 | 464 | 0.98 (0.89, 1.08) | 0.68 | 243,584 | 497 | 1.09 (0.98, 1.21) | 0.11 |
| | Normal➔ Normal | 1,041,296 | 3061 | 1 | | 984,791 | 1434 | 1 | |

Age, body mass index, and economic status were continuous variables.

Adjusted HRs and CIs are derived from Cox proportional regression analysis. All statistical tests are two-sided.

Low HDL-C refers to <40 mg/dL in men and <50 mg/dL in women. Normal HDL-C means ≥40 mg/dL in men and ≥50 mg/dL in women.

CI confidence interval, HDL-C high-density lipoprotein cholesterol, HR hazard ratio.

[a]Adjusted for age, sex, economic status, body mass index, hypertension, diabetes, cerebrovascular disease, heart disease, smoking status, drinking status, physical activity, use of lipid lowering drug, LDL, TG, and liver factors (chronic liver disease, chronic hepatitis B, chronic hepatitis C, and liver cirrhosis).

[b]Adjusted for age, economic status, body mass index, hypertension, diabetes, cerebrovascular disease, heart disease, smoking status, drinking status, physical activity, LDL, TG, and use of lipid lowering drug.

**Table 4 | Effect of HDL-C change by smoking status (adjusted analysis)**

| | | Never smoker | | | | Past/current smoker | | | |
|---|---|---|---|---|---|---|---|---|---|
| | | Total number | Number of cancer | HR (95% CI)[b] | P value | Total number | Number of cancer | HR (95% CI)[b] | P value |
| Gastric cancer | Low → Low | 315,997 | 2458 | 1.05 (1.00, 1.10) | 0.05 | 75,545 | 1595 | 1.08 (1.03, 1.15) | <0.01 |
| | Low → Normal | 243,664 | 2982 | 1.06 (1.02, 1.10) | 0.01 | 82,004 | 2187 | 1.04 (0.99, 1.10) | 0.14 |
| | Normal → Low | 279,714 | | | | 104,999 | | 1.06 (1.02, 1.12) | 0.01 |
| | Normal→ Normal | 1,285,447 | 12,759 | 1 | | 734,884 | 13,871 | 1 | |
| Liver cancer[a] | Low → Low | 315,997 | 1723 | 1.06 (1.00, 1.12) | 0.07 | 75,545 | 858 | 1.13 (1.05, 1.22) | <0.01 |
| | Low → Normal | 243,664 | 1189 | 0.98 (0.91, 1.04) | 0.45 | 82,004 | 837 | 1.03 (0.95, 1.11) | 0.48 |
| | Normal → Low | 279,714 | 1735 | 1.17 (1.11, 1.24) | <0.01 | 104,999 | 1397 | 1.30 (1.22, 1.38) | <0.01 |
| | Normal→ Normal | 1,285,447 | 6554 | 1 | | 734,884 | 6960 | 1 | |
| Colorectal cancer | Low → Low | 315,997 | 3509 | 1.05 (1.01, 1.09) | 0.02 | 75,545 | 1116 | 1.04 (0.97, 1.11) | 0.27 |
| | Low → Normal | 243,664 | 2458 | 1.00 (0.95, 1.04) | 0.93 | 82,004 | 1117 | 0.97 (0.91, 1.03) | 0.34 |
| | Normal → Low | 279,714 | 3177 | 1.09 (1.05, 1.13) | <0.01 | 104,999 | 1734 | 1.11 (1.05, 1.17) | <0.01 |
| | Normal→ Normal | 1,285,447 | 12,320 | 1 | | 734,884 | 10,203 | 1 | |

Age, body mass index, and economic status were continuous variables.

Low HDL-C refers to <40 mg/dL in men and <50 mg/dL in women. Normal HDL-C means ≥40 mg/dL in men and ≥50 mg/dL in women. Adjusted HRs and CIs are derived from Cox proportional regression analysis. All statistical tests are two-sided.

CI confidence interval, HDL-C high density lipoprotein cholesterol, HR hazard ratio.

[a]Adjusted for age, sex, economic status, body mass index, hypertension, diabetes, cerebrovascular disease, heart disease, drinking status, physical activity, use of lipid lowering drug, LDL, TG, and liver factors (chronic liver disease, chronic hepatitis B, chronic hepatitis C, and liver cirrhosis).

[b]Adjusted for age, sex, economic status, body mass index, hypertension, diabetes, cerebrovascular disease, heart disease, drinking status, physical activity, LDL, TG, and use of lipid lowering drug.

group. The impact of longitudinal changes in HDL-C on gastrointestinal cancer risk has not been reported, even though an association between baseline HDL-C level and cancer risk has been reported (Supplementary Table 9). A low baseline HDL-C levels increased the risk of gastric cancer in a cohort study adjusted for *H. pylori* and demographic factors[16] and HDL-C was inversely associated with gastric cancer among postmenopausal women[17]. A Finland study showed no significance between HDL-C and gastric cancer risk[5]. HDL-C was not associated with colorectal cancer in a UK cohort study[8], whereas HDL-C was inversely associated with colorectal cancer in a US Women's Health Study[6] and in European cohort study[18]. This non-constant association between baseline HDL-C and cancer risk may be related to size and portion of study population and study design. However, in the current study, normal HDL-C level at the baseline was maintained in many case (84%) at follow-up, and HDL-C decreased to low levels in small portion (16%). And the effect of these two groups on gastrointestinal cancer risk was far different. Interestingly, low HDL-C persisted just in 55% at follow-up, but HDL-C level increased to the normal range of HDL-C in 45%. It seems that people who have found out that they have low HDL-C through screening are making efforts to reach normal HDL-C. And the persistent low HDL-C increased the risk of gastrointestinal cancer, but low-to-normal group did not increase the cancer risk. Therefore, baseline HDL-C-based cancer risk estimates performed in previous studies have limitations. Furthermore, our findings suggest that persons with low HDL-C at baseline need not be discouraged, and that improving the lipid profile with normal HDL-C can reduce cancer risk to a near extent to people with normal HDL-C at baseline.

The persistent exposure to low HDL-C (HR = 1.17; 95% CI = 1.12–1.23), change from normal to low HDL-C (aHR 1.26; 95% CI = 1.21–1.32), and change from low to normal HDL-C (HR = 1.07; 95% CI = 1.02–1.12) had a higher liver cancer risk compared to the persistently normal HDL-C group. HDL-C level increase by >15 mg/dL among the baseline normal HDL-C group (HR = 1.15; 95% CI = 1.08–1.21) as well as the normal to low HDL-C group (HR = 1.29; 95% CI = 1.24–1.35), increased liver cancer risk compared to the persistently normal group. Our findings suggest that very high HDL-C levels also increase the risk of liver cancer and emphasize that maintaining modest normal range of HDL-C levels is important to reduce the risk of liver cancer. The association between HDL-C level and liver cancer risk remains controversial. HDL-C levels were inversely associated with liver cancer risk in Finland[5] and Swedish cohort studies[10], whereas a Danish cohort study showed no association between HDL-C and liver-biliary cancer[9]. Our finding that maintenance of modest HDL-C level is important to minimize the liver cancer risk, may partly explain the controversial results of previous studies on the association between baseline HDL-C and liver cancer risk.

The change from normal to low HDL-C increased pancreatic and gallbladder cancer risk. Persistently low HDL-C and change from normal to low HDL-C increased the risk of biliary cancer. In analysis after excluding any cancer up to 2 years from baseline, these effects on pancreatic and gallbladder cancer risk were still significant but the effect on biliary cancer risk was not significant. Cohort studies on the association between HDL-C and these cancers have rarely been reported. Two cohort studies showed no association between pancreatic cancer and HDL-C[5,9]. A small case-control study suggested that low HDL-C was associated with gallbladder and bile duct cancers[19].

Our main results were robust to a range of sensitivity analyses for the impact of exposure duration on most gastrointestinal cancers. The effects of HDL-C change on gastrointestinal cancers except biliary cancer were similar in model II (excluding any cancer within 2 years from baseline). The effects of HDL-C change on gastric, colorectal, and liver cancer risk was still significant but the effect on pancreatic, gallbladder, and biliary cancer risk was not significant in model III (excluding any cancer up to 2nd measurement of HDL-C).

**Table 5 | Effect of HDL-C change by age (adjusted analysis)**

| | HDL-C change | <60 yr | | | | ≥60 yr | | | |
|---|---|---|---|---|---|---|---|---|---|
| | | Total number | Number of cancer | HR (95% CI)[b] | P value | Total number | Number of cancer | HR (95% CI)[b] | P value |
| Gastric cancer | Low → Low | 225,422 | 1638 | 1.04 (0.98, 1.10) | 0.17 | 167,110 | 3126 | 1.07 (1.03, 1.12) | 0.001 |
| | Low → Normal | 204,349 | 1572 | 1.03 (0.97, 1.09) | 0.31 | 122,053 | 2489 | 1.04 (0.99, 1.08) | 0.13 |
| | Normal → Low | 234,328 | 1878 | 1.03 (0.98, 1.08) | 0.29 | 151,446 | 3300 | 1.06 (1.02, 1.11) | 0.002 |
| | Normal→ Normal | 1,407,532 | 11,609 | 1 | | 618,555 | 15,084 | 1 | |
| Liver cancer[a] | Low → Low | 225,422 | 932 | 1.21 (1.12, 1.30) | <0.0001 | 167,110 | 1659 | 1.14 (1.07, 1.21) | <0.0001 |
| | Low → Normal | 204,349 | 758 | 1.02 (0.95, 1.11) | 0.57 | 122,053 | 1271 | 1.09 (1.02, 1.17) | 0.01 |
| | Normal → Low | 234,328 | 1266 | 1.32 (1.23, 1.41) | <0.0001 | 151,446 | 1872 | 1.22 (1.16, 1.29) | <0.0001 |
| | Normal→ Normal | 1,407,532 | 6146 | 1 | | 618,555 | 7405 | 1 | |
| Colorectal cancer | Low → Low | 225,422 | 1656 | 1.06 (0.001, 1.12) | 0.04 | 167,110 | 2983 | 1.00 (0.96, 1.05) | 0.96 |
| | Low → Normal | 204,349 | 1356 | 0.95 (0.90, 1.01) | 0.10 | 122,053 | 2224 | 0.98 (0.94, 1.03) | 0.43 |
| | Normal → Low | 234,328 | 1776 | 1.06 (1.01, 1.12) | 0.02 | 151,446 | 3143 | 1.09 (1.05, 1.14) | <0.0001 |
| | Normal→ Normal | 1,407,532 | 9895 | 1 | | 618,555 | 12,674 | 1 | |
| Pancreatic cancer | Low → Low | 225,422 | 526 | 1.11 (1.01, 1.23) | 0.04 | 167,110 | 1120 | 1.03 (0.96, 1.11) | 0.47 |
| | Low → Normal | 204,349 | 460 | 1.11 (1.00, 1.23) | 0.05 | 122,053 | 815 | 1.01 (0.93, 1.09) | 0.87 |
| | Normal → Low | 234,328 | 611 | 1.21 (1.10, 1.32) | <0.0001 | 151,446 | 1049 | 1.05 (0.98, 1.13) | 0.16 |
| | Normal→ Normal | 1,407,532 | 2811 | 1 | | 618,555 | 4140 | 1 | |
| Gallbladder cancer | Low → Low | 225,422 | 168 | 1.14 (0.95, 1.36) | 0.15 | 167,110 | 419 | 0.91 (0.81, 1.03) | 0.12 |
| | Low → Normal | 204,349 | 132 | 1.04 (0.86, 1.27) | 0.67 | 122,053 | 333 | 0.98 (0.87, 1.11) | 0.76 |
| | Normal → Low | 234,328 | 172 | 1.12 (0.94, 1.33) | 0.21 | 151,446 | 462 | 1.12 (1.01, 1.25) | 0.03 |
| | Normal→ Normal | 1,407,532 | 815 | 1 | | 618,555 | 1724 | 1 | |
| Biliary cancer | Low → Low | 225,422 | 208 | 0.95 (0.82, 1.11) | 0.53 | 167,110 | 684 | 0.95 (0.87, 1.04) | 0.27 |
| | Low → Normal | 204,349 | 190 | 0.95 (0.81, 1.11) | 0.53 | 122,053 | 513 | 0.95 (0.86, 1.05) | 0.31 |
| | Normal → Low | 234,328 | 284 | 1.19 (1.04, 1.36) | 0.01 | 151,446 | 677 | 0.98 (0.9, 1.07) | 0.60 |
| | Normal→ Normal | 1,407,532 | 1380 | 1 | | 618,555 | 3115 | 1 | |

Age, body mass index, and economic status were continuous variables.
Low HDL-C refers to <40 mg/dL in men and <50 mg/dL in women. Normal HDL-C means ≥40 mg/dL in men and ≥50 mg/dL in women. Adjusted HRs and CIs are derived from Cox proportional regression analysis. All statistical tests are two-sided.
CI, confidence interval; HDL-C, high density lipoprotein cholesterol; HR, hazard ratio.
[a]Adjusted for age, sex, economic status, body mass index, hypertension, diabetes, cerebrovascular disease, heart disease, drinking status, smoking status, physical activity, use of lipid lowering drug, LDL, TG, and liver factors (chronic liver disease, chronic hepatitis B, chronic hepatitis C, and liver cirrhosis).
[b]Adjusted for sex, economic status, body mass index, hypertension, diabetes, cerebrovascular disease, heart disease, drinking status, smoking status, physical activity, LDL, TG, and use of lipid lowering drug.

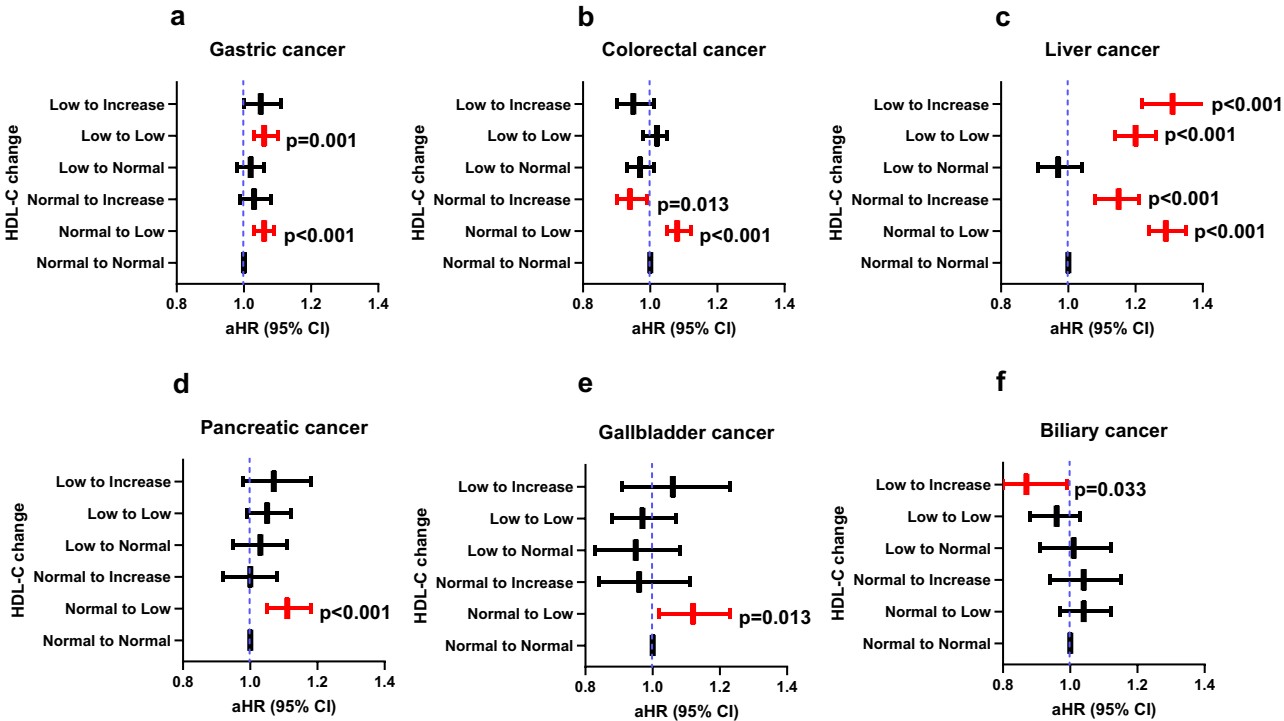

**Fig. 3 | The effect of further categorization of HDL-C change on cancer risk.**
Adjusted hazard ratios (aHR) for gastric (**a**), colorectal (**b**), liver (**c**), pancreatic (**d**), gallbladder (**e**), and biliary cancers (**f**). ΔHDL-C = [(HDL-C at follow-up) − (HDL-C at baseline)]. *X*-axis is adjusted HR and *y*-axis is HDL-C change group. Adjusted HRs and CIs are derived from Cox proportional regression analysis. The dashed blue lines illustrate an HR of 1. The reference is a persistently normal (Normal-to-Normal) group. The HR is indicated with the central bar and the error bars show the 95% confidence interval (CI). Red bar is statistical significant HR with CI. Low to Increase (*n* = 125,709), Low to Low (*n* = 387,739), Low to Normal (*n* = 205,486), Normal to Increase (*n* = 194,998), Normal to Low (*n* = 385,774), and Normal to Normal (*n* = 1,831,089). The exact hazard ratios and 95% confidence intervals are provided in Supplementary Table 7. HDL-C high density lipoprotein cholesterol.

Effect modifications by sex, smoking status, and age group were observed in subgroup analysis. Normal-to-low HDL-C change increased the risk of gastric, colorectal, liver, and pancreatic cancer in both men and women, whereas persistent low HDL-C increased the risk of those cancers only in women but not in men. In the subgroup analysis by smoking status, the hazardous effect of normal-to-low HDL-C change on gastric, colorectal, and liver cancer was significant in both never-smokers and ever-smokers, whereas persistent low HDL-C increased the risk of gastric and colorectal cancers only in never smokers and the risk of liver cancer only in ever-smokers. In subgroup analysis by combination of sex and smoking status (men-smoker, men-never smoker, women-smoker, women-never smoker), the effect of HDL-C change on women's gastric cancer risk was significant only in never smokers and the effect of HDL-C change on men's gastric cancer risk was significant only in smokers. These patterns were observed in pancreatic cancer. The hazardous effect of persistent low or normal-to-low HDL-C on liver cancer risk was observed in all 4 groups, but effect size was markedly high in smokers regardless of sex. The mechanism of this discrepant effect of HDL-C change according to sex and smoking status in several cancers is unknown. A plausible explanation is that the immunologic and inflammatory response to smoking further promotes a low HDL-C level−related cancer risk in current smokers. The related mechanisms need to be investigated in the future. Previous epidemiologic studies have reported sex-discrepant effects of BMI on cancer risk[20]. The interaction between BMI and smoking status on cancer risk has also been previously reported[20,21]. In a previous small study (259 cases of lung cancer in 14,547 members of atherosclerotic risk) using binary HDL-C, low HDL-C at baseline was associated with lung cancer only in past smokers (HR = 1.77; 95% CI = 1.05−2.97)[22].

Interestingly, subgroup analysis by age group showed a discrepant effect on individual gastrointestinal cancers. The effect of HDL-C change on the risk of gastric and gallbladder cancer was significant only in age ≥60 years. The effect of HDL-C change on the risk of pancreatic and biliary cancer was significant only in age <60 years. The effect of HDL-C change on the risk of liver and colorectal cancer risk was significant in both age groups. The mechanism of site specific age discrepancy in the association between HDL-C change and digestive cancer risk is unknown. A different peak onset ages and site-specific biologic response to low HDL-C may partially contribute to this site specific age discrepancy in the association between HDL-C changes and gastrointestinal cancer risk.

We also investigated the effect of absolute HDL-C change on cancer risk. A decrement of absolute HDL-C level (ΔHDL-C < −10 mg/dL) increased the risk of gastric, colorectal, pancreatic, gallbladder, and biliary cancers comparing to stable HDL-C group. An increment of HDL-C from baseline (Δ HDL-C ≥ 25 mg/dL) was also associated with an increased risk of gastric (7%) and liver cancer (41%), whereas an increment of HDL-C (Δ HDL-C = 5−25 mg/dL) slightly reduced the risk of colorectal cancer (6%). These results emphasize the importance of avoiding a decrement of HDL-C to prevent all digestive cancers and encourage an increment of HDL-C to prevent colorectal cancer.

The favorable effect of HDL-C on cancer appears to be related to its anti-inflammatory and anti-oxidative properties[12,23]. LDL-C in the tumor microenvironment can promote pro-oxidant and pro-inflammatory activities, whereas cholesterol and its metabolites can be removed from cancer cells by HDL-C[24]. HDL-associated apolipo-proteins and ATP-binding cassette transporters might have anti-tumorigenic effects[25,26]. HDL-C also has anti-apoptotic effects[27]. This anti-apoptotic effect of HDL-C may partially explain the increased risk of liver cancer in the change from the normal to the high HDL-C group. Liver is a complex functional organ related to lipid metabolism and is one of organs of estrogen action[28]. Functional investigation for the link of HDL-C and liver cancer may clarify our epidemiologic findings.

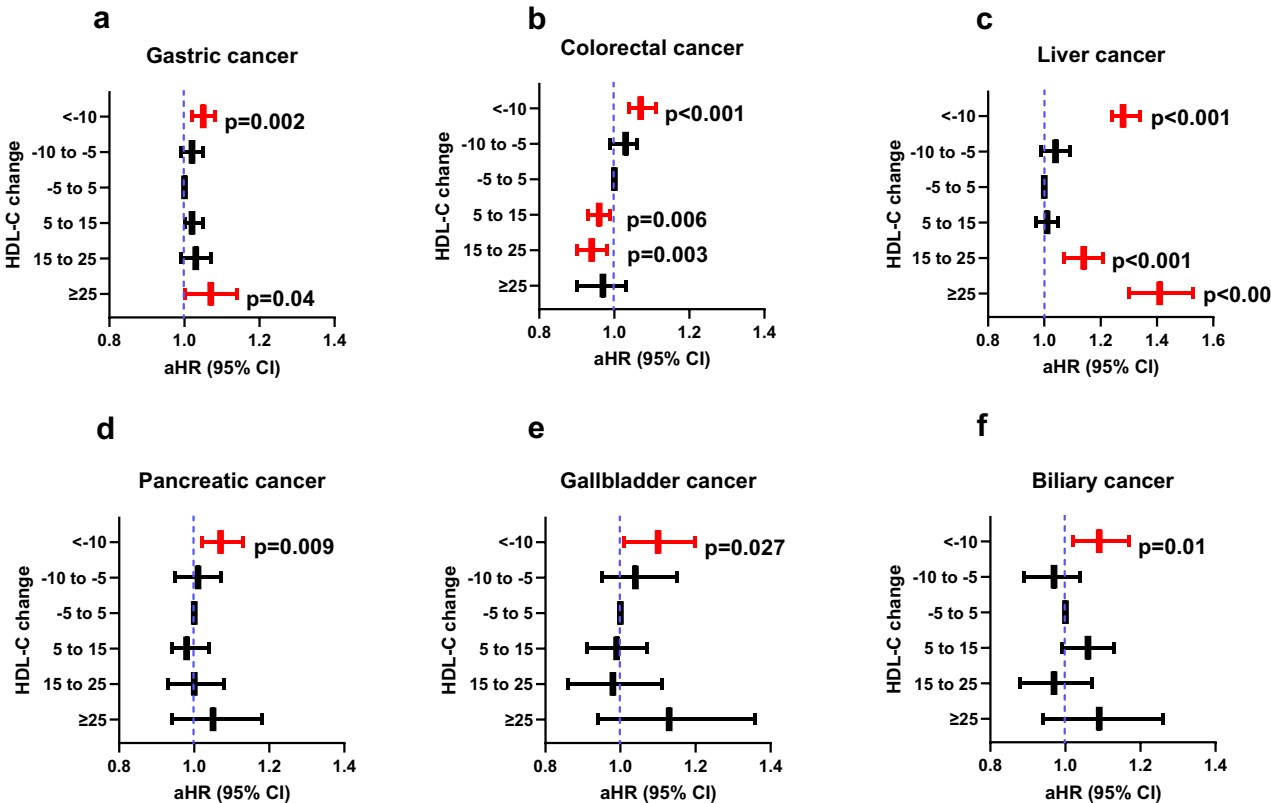

**Fig. 4 | The effect of absolute HDL-C change on cancer risk.** Adjusted hazard ratios (aHR) for gastric (**a**), colorectal (**b**), liver (**c**), pancreatic (**d**), gallbladder (**e**), and biliary cancers (**f**). ΔHDL-C = [(HDL-C at follow-up) − (HDL-C at baseline)]. *X*-axis is adjusted HR and *y*-axis is HDL-C change group. Adjusted HRs and CIs are derived from Cox proportional regression analysis. The dashed blue lines illustrate an HR of 1. The reference is a stable group. The HR is indicated with the central bar and the error bars show the 95% confidence interval (CI). Red bar is statistical significant HR with CI. ΔHDL < −10 (n = 561,153), ΔHDL = −10−−5 (n = 424,815), ΔHDL = −5−5 (n = 1,121,127), ΔHDL = 5−15 (n = 702993), ΔHDL = 15−25 (n = 234,723), ΔHDL ≥ 25 (n = 85,984). The exact hazard ratios and 95% confidence intervals are provided in Supplementary Table 8. HDL-C high-density lipoprotein cholesterol.

However, a previous study suggests that high cholesterol consumption by malignant cells to promote tumor growth might deplete plasma cholesterol and HDL-C levels[29]. Therefore, low HDL-C might be the epiphenomenon of cancer-related inflammation[30,31]. To avoid this issue, we enrolled a cancer-screened population and excluded patients diagnosed with any cancer before baseline and at baseline and excluded persons who diagnosed with 1 years and 2 years after baseline enrollment (Model I and Model II). And we further excluded those who were diagnosed with any cancer until the second measurement of HDL-C (model III).

Associations between gastrointestinal cancer risk and other lipids such as total cholesterol, LDL-C, triglyceride have also been studied[5,6,8,10]. A cohort study from Finland suggested that HDL-C was inversely associated with the risk of several site cancers but total cholesterol had no association with cancer risk during long term follow-up[5]. A US women cohort study showed that HDL-C was inversely associated with colorectal and lung cancer but LDL-C was not significantly associated with risk of total cancer or any site-specific cancers[6]. LDL and triglycerides are affected not only by long-term exercise, diet, and obesity, but also by short-term lipid-lowering drugs in case of LDL-C[32], and by recent diet and alcohol consumption in case of triglycerides. On the other hand, HDL-C is influenced by long term exercise and diet pattern rather than short-term diet or lipid lowering drugs. This relative stability of HDL-C may be used as a marker for cancer risk assessment, so we focused on HDL-C. Actually our previous studies showed no association between gastric cancer and other lipids (total cholesterol, triglyceride, and LDL)[16,33]. The impact of HDL-C change on most gastrointestinal cancer in the adjusted analysis that did not consider LDL and TG was similar to the results in the adjusted analysis that included LDL-C and TG. However, the effect of HDL-C change on biliary tract cancer risk was significant in adjusted analyzes including LDL-C and TG, but not significant in adjusted analyzes excluding LDL-C and TG. These results suggested that HDL-C change affect gastrointestinal cancer risk regardless of other lipid level.

This study has considerable strengths. First, this is the first study to investigate the association between HDL-C changes and cancer risk of many gastrointestinal sites. A large database allowed us to investigate the effects of HDL-C changes on individual cancer risks. Furthermore, our results were robust to a range of sensitivity analyses for the impact of exposure duration on most cancers. Second, we showed a sex discrepancy in the association between HDL-C level change and individual digestive cancer risk. Furthermore, the hazardous effect of HDL-C changes on individual cancers differed according to smoking status. Subgroup analysis by combination of sex and smoking status clarified their association in the effect of HDL-C on cancer risk. Third, we also showed effect modification by age group in the impact of HDL-C change on digestive cancer risk. Fourth, we also explored that a decrement of absolute HDL-C level (ΔHDL-C < −10 mg/dL) increased the risk of individual gastrointestinal cancers but marked increase of HDL-C had a site specific discrepant effect. Fifth, we used high-quality data from the NHISS, including direct measurement of lipids and BMI, detailed information on many covariates, and nearly complete sensing of cancer detection. Many previous studies have used data from the NHISS and validated their quality[13,14]. Finally, this national health examination cohort represents the general Korean population aged ≥40 years.

Nevertheless, this study has several limitations. The median follow-up period was 8.4 years. Although this duration may be relatively short to investigate cancer risk, the large cohort size (23,389,010 person years and enough number of events) may have partially compensated for this limitation. Second, we selected individuals who underwent two consecutive general health examinations, and follow-up eligible individuals comprised 63% of the baseline eligible individuals. Finally, no external validation was performed in this study. However, this study will promote similar research in other countries.

In conclusion, persistent exposure to low HDL-C and HDL-C level changes from normal to low levels increased overall and most digestive cancer risks compared to persistently normal HDL-C. These effects of HDL-C changes were robust to a range of sensitivity analyses. The risk of individual digestive cancer according to HDL-C change differs on the level of sex and smoking status. This study emphasizes the importance of maintaining normal HDL-C levels to prevent cancer development, especially in ever-smokers.

## Methods
### Data extraction and contents of data
This study was approved by the Institutional Review Board of Kyungpook National University Hospital, Chilgok (KNUHC 2017-12-022). In this large population-based cohort study, we used data from the Korean NHISS (REQ202204452-004), which cover approximately 98% of the Korean population aged ≥40 years every 2 years. We gathered the Base One Foundation Component Library (BFC), T20, the National General Health Examination (NGHE), and national cancer screening from NHISS.

We extracted sex and economic status from BFC; the disease codes of the International Classification of Disease, 10th revision (ICD 10) from T20 disease code; and demographic and laboratory data from the NGHE. Sex in this study is a biological sex registered in NHISS. The questionnaires included chronic diseases such as diabetes mellitus, hypertension, cerebrovascular disease, and ischemic heart disease, medication (anti-hypertensive drug, ant-diabetes drug, drug for heart disease, antiplatelet or anticoagulant drug, lipid lowering drug, etc), smoking status, alcohol consumption frequency, physical activity (frequency per week), and family history. Questionnaires also included liver-related variables (any liver disease history, chronic hepatitis B, chronic hepatitis C, and liver cirrhosis). Laboratory data included body mass index (BMI, weight/height$^2$ [kg/m$^2$]), blood pressure, lipid and fasting glucose levels. Weight and height were directly measured on the day of the health examination and blood samples were collected after 12 h of fasting. Fasting serum glucose and lipid levels were measured in units of milligram/deciliter (mg/dL) using fresh serum in each screening center. Korean national general health screening have provided glucose and lipids examination to all adults over than 40 years every 2 years during study periods. Therefore, missing rate of glucose and lipid is extremely low in the study period. Age was defined as the age at the time of national general health examination. Economic status was extracted as household income twentile. Smoking status was categorized with never, past, or current smoker. Alcohol consumption frequency was classified as none, 1/week, 2–3/week, 4–5/ week, and ≥6/week. Moderate physical activity refers to "walking or exercising and feeling mild dyspnea for more than 30 min per day."

### Handling of missing data

1. *Missing data for hypertension and diabetes mellitus* in self-reported questionnaires was replaced using medication, blood pressure, and fasting glucose extracted from laboratory data in general health examination. Among missing data of hypertension or diabetes mellitus, use of antihypertensive or anti-diabetes drug was defined as presence of hypertension or diabetes. Among missing data of hypertension, systolic blood pressure ≥140 mmHg

or diastolic blood pressure ≥90 mmHg was defined as hypertension. Among missing data of diabetes mellitus, fasting glucose ≥126 mg/dL was defined as presence of diabetes.
2. *Missing data for ischemic heart disease (angina or myocardial infarction) and cerebrovascular disease* was replaced using T20 disease code within 1 year.

   T20 disease code includes all ICD10 code registered whenever patients visit any medical institute. Ischemic heart disease includes I20–I25. Cerebrovascular disease includes I60–I69.
3. *Other categorical data*: Missing data was put as blank or unknown.
4. *Continuous variables such as age, body mass index, and lipid*: Missing data was put as blank. After reasonable correction of missing data, we handled the missing data based on the missing rate. If variables have low missing rate (<2%) among the final eligible population, we deleted missing data list-wise in the adjusted analysis. If variables have considerable missing rate, we set the missing data as unknown group in adjusted analysis.

### Baseline enrollment and follow-up
From 2009 to 2017, national general health examination (NGHE) provided measurement of HDL-C to all Korean persons aged 40 years or older every two years. Among 6.18 million who underwent both NGHE and any cancer screening from January to December 2010, patients with any pre-existing cancer and persons who did not undergo gastric cancer screening were excluded. Among cancer free individuals who underwent both NGHE and National gastric cancer screening from January to December 2010 (4.373 million), we excluded persons with any C-codes or death within 12 months from index month (the month of health examination) ($n = 49,991$) (Fig. 1a). Death data from the National Statistical Office were also provided by the National Health Insurance Service. Nearly complete sensing of new cancer is possible through extracting C code from NHISS. In order to receive a special exemption for cancer, the cancer code should be registered with diagnostic evidence of cancer such as pathologic results at hospital. If any cancer code is registered, the patients pay just 5% for cancer related medical services. Therefore, cancer (C) code extracted from NHISS is highly reliable. Individuals who did not undergo NGHE in 2014 or absence of HDL-C values in 2014 ($n = 1,188,176$) and unknown sex type ($n = 3991$) were excluded [model I; $n = 3,130,795$, Fig. 1b]. All cancer codes (C codes) were extracted up to December 2021. Common gastrointestinal cancers included gastric (C16), liver (C22), colorectal (C18, C19, C20), pancreatic (C25), gallbladder (C23), and biliary (C24) cancers. Nearly complete sensing of new cancer is possible through extracting C code from NHISS. The NHISS provided raw data after eliminating personal identification information and this study did not affect the disease course. Therefore informed consent is waived in analysis using NHISS data under NHISS regulation.

### Statistics and reproducibility
**1) Determination of sample size**. We did not estimate the sample size because the number of samples which could be used was fixed. However, we have provided calculations for the primary analysis with a specified power.

**Framework**. Persistent low HDL will increase the risk of individual cancer risk comparing to persistent normal HDL group. Change from normal HDL-C to low HDL-C will increase the risk of individual cancer risk comparing to persistent normal HDL group. We put Alpha error as 0.05 and Power as 80%.

**Minimum sample size to evaluate the risk of colorectal cancer**. In two independent groups, we put the colorectal cancer incidence of Group 1 and Group 2 is 0.6% and 0.5, respectively. The sample size is 171,724. In this study, HDL-C was categorized as four different groups. Therefore minimal sample size = 171,724 × 2 = 343,448.

**2) HDL-C change group.** HDL-C levels were extracted at baseline (2010) and follow-up (2014). Changes in HDL-C levels were classified into four groups: persistent normal (normal-to-normal), change from normal to low (normal-to-low), change from low to normal (low-to-normal), and persistent low (low-to-low) (Fig. 1c). According to the Adult Treatment Panel III (ATP III)[15], normal HDL-C levels were defined as HDL-C ≥ 40 mg/dL in men and ≥50 mg/dL in women. Low HDL-C levels were defined as HDL-C < 40 mg/dL in men and <50 mg/dL in women. We also classified HDL-C change using continuous value. Change of HDL-C [ΔHDL-C = (HDL-C at follow-up) − (HDL-C baseline)] were classified as decrease (ΔHDL-C < −10 mg/dL, ΔHDL-C = −5−−10 mg/dL), stable (ΔHDL-C = −5–5 mg/dL), increase (ΔHDL-C = 5–15 mg/dL, ΔHDL-C = 15 ~ 25 mg/dL, and ΔHDL-C ≥ 25 mg/dL). We further categorized HDL-C change using ATP III and absolute HDL-C change [ΔHDL-C = (HDL-C at follow-up) − (HDL-C baseline)] to investigate which further increment of HDL-C among baseline normal HDL-C can affect the risk of gastrointestinal cancer (Fig. 1c). Persistent normal group was further classified into persistent normal and normal-increase (baseline normal and increase ≥15 mg/dL at follow-up from baseline) groups. Baseline low group was categorized into persistent low, low-to- normal, and low-to-increase.

**3) Statistical analysis.** Statistics for the variables are presented as numbers (percentages) for categorical variables and as means (standard deviations) or medians (interquartile ranges) for continuous variables. First, we investigated the risk of gastrointestinal cancer according to baseline HDL-C. We calculated person-years from the date of baseline national general health examination to the following censoring events, whichever occurred first: occurrence of gastrointestinal cancer, death, or end of the study (December 31, 2021).

Cancer risk according to HDL-C change was measured with hazard ratios (HRs) and 95% confidence intervals (CIs) using Cox proportional regression analysis. The persistently normal group was set as the reference group. The association between covariates and cancer risk was assessed using the Cox regression analysis. We performed a multivariate analysis adjusted for significant confounders, such as age, sex, BMI, smoking status, alcohol intake, economic status, use of lipid lowering drugs, hypertension, diabetes, cerebrovascular disease, heart disease, low density lipoprotein cholesterol (LDL-C), triglyceride, and physical activity. Liver cancer analysis was additionally adjusted for liver factors such as liver disease, chronic hepatitis B, chronic hepatitis C, and liver cirrhosis. Among the final eligible population, the highest missing rate was for economic status (1.7%). Therefore, we deleted missing data list-wise in the adjusted analysis.

To conduct sensitivity analysis for the impact of exposure duration, we excluded cancers that developed within 2 year from the first measurement of HDL-C level and then analyzed the impact of HDL-C change on each cancer risk (*model II*). And persons who had been diagnosed with cancer up to second measurement of HDL-C (2014) were further excluded (*model III*) (Fig. 1b). We conducted an interaction analysis (joint test) between the well-known important cofactors (sex and smoking status) and HDL-C level changes in the cancer risk. And then, a subgroup analysis was performed based on sex and smoking status. All analyses were performed using the SAS software (version 9.4; SAS Institute, Cary, NC, USA). All statistical tests were two-sided, and values of $P < 0.05$ were considered statistically significant.

### Reporting summary
Further information on research design is available in the Nature Portfolio Reporting Summary linked to this article.

## Data availability
Raw data are available offline and not online because analysis can be conducted only in a closed office regulated by National Health Insurance Service System (NHISS). If someone want to use the raw data, they have to request to NHISS via homepage (https://nhiss.nhis.or.kr/bd/ay/bdaya001iv.do). Institutional IRB, research proposal documents, data extraction plan, personal information use agreement of researchers, and compliance Statement should be submitted on the NHISS homepage. After NHISS approve the use of data, they can analyze the data in a closed office regulated by NHISS. It takes about 1 years; 1) Approval of Institutional IRB (2 months), 2) Approval of data use from NHISS (6–8 months), and 3) Analysis room assignment (2–3 months). Analysis data set number used in this study is REQ202204452-004. This data set included persons who underwent general health examination and cancer screening at 2010 and follow up general heath examination at 2014. The data also included ICD-10 code for any cancer and death data from 2010 to 2021. This data set will be expired at April 2025.

## Code availability
Analysis code (SAS code) is available in Github (*namsy2021 / HDL-C-change-GI-cancer-analysis-code*). https://github.com/namsy2021/HDL-C-change-GI-cancer-analysis-code.

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

## Acknowledgements

This work was supported by the National Research Foundation, Republic of Korea (NRF-2019R1F1A1063990 and NRF-2022R1A2C2013044) (S.Y. Nam). The funders had no role in the design and conduct of the study, including collection, management, analysis, and interpretation of the data, and preparation, review, or approval of the manuscript.

## Author contributions

Concepts, design, grant, and visualization: S.Y. Nam. Data management: S.Y. Nam and J.W. Jo. Data analysis and interpretation: S.Y. Nam, J.W. Jo, and C.M. Cho. Writing a draft: S.Y. Nam. Critical revision: S.Y. Nam, J.W. Jo, and C.M. Cho. Responsibility for data accuracy and decision to submit for publication: S.Y. Nam. All authors wrote the manuscript and approved the final approval.

## Competing interests

The authors declare no competing interests.
