## [Peer Review File · Nature Communications]

REVIEWER COMMENTS

Reviewer #1 (Remarks to the Author):

The authors report results from a very large Korean study on HDL-cholesterol change and risk of GI tract cancers. The dataset is impressive and the analysis provides novel information. However, I have a few suggestions that could improve/guide the interpretation.

1. The authors have measured HDL twice in 2009 and 2013 and have performed HDL change analyses for incident cancers developing after 2013 (until 2017). I would be interested to see a full-blown longitudinal analysis (exploring also potential non-linearity) on top of the HDL change analysis that could also use cases diagnosed between 2009 and 2013.
2. The most important limitation is the lack of mutual adjustment for other lipoproteins. As the different cholesterol particles are correlated, it is not clear if the results are due to HDL or other lipoproteins.
3. Introduction, the authors report briefly on the association of HDL with CVD, but this association has been challenged on whether it is causal or not, and other lipoproteins seem more important.
4. Methods, it is not clear why the study started in 2009 and not earlier.
5. Methods, the authors reported "Patients who had been diagnosed with cancer before 2009 and within 1 year of baseline enrollment, and those who died within 1 year of enrollment were excluded" It is unclear how prevalent cancers were assessed and how complete this assessment is.

Reviewer #2 (Remarks to the Author):

Re: Impact of Longitudinal Change of High-density Lipoprotein Cholesterol on Gastrointestinal Cancer Risk in a Large Cohort

The authors have explored the association between changes in HDL-C and risk of gastrointestinal cancer, using a cohort of 2.8 million cancer-free individuals followed for 8 years. This is a great resource, and this does add novel findings on the impact of HDL-c on cancer risk. For the most part the paper is well written. I do recommend two additional analyses for improvement of this study: (1) show results by both sex and smoking status (i.e., female smokers, female nonsmokers, male smokers etc.) and (2) Incorporate baseline HDL-c level in the analysis of HDL-c continuous change. I recommend supplying a supplementary table that shows how excluded participants (those without a second measurement) differed from the individuals included in the paper. The discussion could be improved upon by focusing less on repeating direct results and more on interpretation of findings. Specific comments are provided for each section below:

Abstract

1. Please include a sentence about the methods (time to event analysis, HR calculation, adjustment factors). Note that you have separated smokers and non-smokers as well, so that the conclusions follow.

Introduction

1. In the second paragraph it would be worthwhile to include a reason for investigating the change in HDL-C levels- i.e., what evidence is there that suggests the change over time may be important for risk? I believe more can be added to the introduction to make a case for this study.
2. The last few sentences of the introduction are methods focused and could be abbreviated.

Results

1. Subgroup analysis among baseline normal HDL-C groups is hard to follow. Why was an increase in HDL-C tested for individuals who were in the normal-to-low group? None of these patients should have an increase in HDL-c correct? Please clarify this in the results and in the methods sections.
2. Subgroup analysis by sex: Please specify what change you are referring to in the following sentences: "A change in the normal to low HDL-C group was associated with an increased risk of

liver cancer in men (aHR 1.20; 95% CI 1.06-1.36) and a change in the low to normal HDL-C group was associated with a reduction in women's liver cancer risk (aHR 1.32 0.83; 95% CI 0.69-0.99).
"A change in the normal to low HDL-C group was associated with an increased risk of pancreatic cancer in women (aHR 1.19; 95% CI 1.02-1.39) and had a borderline association with women's gallbladder cancer risk (aHR 1.25; 95% CI 0.97-1.62)".

a. Is it meant to state that "Those in the normal- to-low HDL-C group had an increased risk of liver cancer...". If that is the case, please reword; as it is stated it seems there is an additional change involved here.

3. Effect of absolute HDL-C change on cancer risk: The following statement is unclear, please reword; I believe instead of increment, you meant "increase":

a. "An increment of HDL-C at follow-up was associated the reduced risk of gastric cancer (HR, 0.96 for Δ HDL C 5-15 mg/dL and 0.94 for Δ HDL-C 15-25 mg/dL) (Fig. 4B)."

4. In Extended Data Table 2, "HDL-G" is written – should be "HDL-C"

5. Given the interactions of both sex and smoking seen in this study, I recommend reporting results broken down by sex- and smoking status together.

6. For figure 4, where change in HDL is measured continuously, it would be advantageous to break this down further based on initial HDL-C group. i.e., report the results for change seen among individuals with (a) normal HDL-c at baseline (b) low HDL-c at baseline, and (c) high HDL-c at baseline.

Discussion

1. The discussion is a bit choppy and often repeats results without adding relevant context, e.g.: "The hazardous effect of persistent low and change from normal to low HDL-C on liver cancer was significant only in smokers." – This statement does not explain the role of smoking in the association between HDL-c change and cancer risk and simply repeats results. It is not linked with the discussion on smoking below (lines 209-217), which would improve the flow.

2. I believe the flow and the quality of the discussion can be improved by focusing more on potential explanations for findings and previous literature and by limiting repetition of results. For example, focus on what results agree or disagree with previous literature, without restating all the results. Lines 224-231 add interesting insight into mechanisms and could be expanded upon.

Methods

1. Given the exclusion of individuals who did not have a second measurement (undergo an examination in 2013), I believe it needs to be clearly stated throughout the paper that this study only applies to individuals who have been followed (remained cancer-free) for at least four years. This speaks to the generalizability of the findings, though it was not made clear throughout interpretation of results. Further, was risk of cancer examined among those individuals who did not undergo a health examination in 2013? It would be of interest to know how these individuals compared with those who were included in the study (e.g., comorbidities, initial HDL-C levels, demographics, cancer outcomes) to understand better who is excluded. (also mentioned in my comments on "Results" above)

2. For analysis of continuous change in HDL, the baseline HDL-c should be accounted for; I recommend stratifying and reporting the results for change seen among individuals with (a) normal HDL-c at baseline (b) low HDL-c at baseline, and (c) high HDL-c at baseline. (Also mentioned in my comments on "Results above")

3. In the following sentence physical activity is listed twice: "We performed a multivariate analysis adjusted for significant confounders, such as age, sex, BMI, smoking status, alcohol intake, physical activity, economic status, use of lipid lowering drugs, hypertension, diabetes, cerebrovascular disease, heart disease, and physical activity"

4. Please list the endpoints/censorship variables in the section "Statistical Analysis"

RESPONSE TO REVIEWERS' COMMENTS

Thank you so much for giving the opportunity of the revision. Authors appreciate the critical and thoughtful review comments. We meticulously revised the manuscripts according to reviewers' comments and included point-by-point responses to each comment, stating clearly how the text has been changed. We indicated precisely where changes have been made in the manuscript, by changing the font to red color.

REVIEWER COMMENTS

Reviewer #1 (Remarks to the Author):

The authors report results from a very large Korean study on HDL-cholesterol change and risk of GI tract cancers. The dataset is impressive and the analysis provides novel information. However, I have a few suggestions that could improve/guide the interpretation.

1. The authors have measured HDL twice in 2009 and 2013 and have performed HDL change analyses for incident cancers developing after 2013 (until 2017). I would be interested to see a full-blown longitudinal analysis (exploring also potential non-linearity) on top of the HDL change analysis that could also use cases diagnosed between 2009 and 2013.

Answer> Thank you for the important comment. We tried several models to investigate the effect of HDL-C change on GI cancer risk. We firstly analyzed the effect of HDL-C on GI cancer risk using cancer cases diagnosed 1 year (2010) after baseline measurement of HDL-C (2009) [model III]. However, we think that this model is not enough to avoid the possibility of epiphenomenon. A previous study suggests that high cholesterol consumption by malignant cells to promote tumor growth might deplete plasma cholesterol and HDL-C levels.²⁹ Therefore, low HDL-C might be the epiphenomenon of cancer-related inflammation.^{30, 31} To avoid this issue, we further excluded patients diagnosed with any cancer until the second measurement of HDL-C (main model, model I). In addition, we conducted sensitivity analysis to evaluate reverse causation after excluding individuals who were diagnosed with any cancer until 1 years after the second measurement of HDL-C (Model II). We focused on main model.

We provided mode III analysis in **Supplementary Table 6**. To analyze model III meticulously, we need raw data set. However, raw data was expired during waiting for review. It takes 1 year

to get the same raw data again from NHISS.

In a previous study (2020, JNCI), follow-up time began after the last weight assessment and ended at the date of incident invasive breast cancer diagnosis unless the women were first censored for death, loss to follow-up, or administrative end of follow-up to investigate the effect of BMI change on breast cancer risk (similar to our model I).

(J Natl Cancer Inst. 2020 Sep; 112(9): 929–937.)

We also conducted several categorization of HDL-C change.

Categorization A: 4 groups by ATP III (Fig 2)

Categorization B: 6 groups using ATP III and Δ HDL-C (revised Fig 4)

Categorization C: absolute HDL-C change [Δ HDL-C = (HDL-C at 2013) - (HDL-C at 2009)] (revised Fig 5)

We added model III and categorization B in the Method, Results, Discussion, Tables (Extended and Supplementary Table), and Figures.

(Methods)

Baseline enrollment and follow-up (p25)

From 2009 to 2017, NGHE provided measurement of HDL-C to all Korean persons aged 40 years or older. NHISS provided raw data after excluding persons with gastrointestinal cancer from 2004 to 2007. Subjects who underwent NGHE (4.983 million) and cancer screening (4.75 million) in 2009 and NGHE in 2013 were included in this study. Among subjects who underwent both NGHE and National Cancer Screening from January to December 2009 (4.614million), we excluded persons with any C-codes within 12 months from index month

(the month of health examination) (n=65,791) and the persons who died within 1 year of enrollment (n=14,863) (**Fig. 1A**). Death data from the National Statistical Office were also provided by the National Health Insurance Service. Nearly complete sensing of new cancer is possible through extracting C code form NHISS. In order to receive a special exemption for cancer, the cancer code should be registered with diagnostic evidence of cancer such as pathologic results at hospital. If any cancer code is registered, the patients pay just 5% for almost medical services. Therefore, cancer (C) code extracted from NHISS is highly reliable. Individuals who did not undergo NGHE in 2013 (n= 1,665,999) or absence of HDL-C values in 2013 were excluded [**model III; n=2,865,902**]. And persons who had been diagnosed with cancer (n=68,029) up to 2013 were further excluded [**model I, main model; n=2,797,873**] (**Fig 1B**).

Statistical analysis (p26)

We further categorized HDL-C change using ATP III and absolute HDL-C change [$\Delta\text{HDL-C} = (\text{HDL-C at follow-up}) - (\text{HDL-C baseline})$] (**Fig. 1C and supplementary material p5**).

Persistent normal group was further classified into persistent normal and normal-increase (baseline normal and increase ≥ 15 mg/dL at follow-up from baseline) groups. Baseline low group was categorized into persistent low, low-to- normal, and low-to-increase.

(Supplementary Material p5)

Change of HDL-C was classified into 6 groups using APTII and $\Delta\text{HDL-C}$

- ◆ Persistent normal HDL-C (normal HDL-C at 2009 and normal HDL-C at 2013 and $\Delta\text{HDL-C} < 15\text{mg/dL}$)
- ◆ Conversion of normal to low HDL-C
- ◆ Normal to increase HDL-C (normal HDL-C at 2009 and $\Delta\text{HDL-C} \geq 15\text{mg/dL}$)
- ◆ Persistent low (low HDL-C at 2009 and low HDL-C at 2013 and $\Delta\text{HDL-C} < 15\text{mg/dL}$)
- ◆ Conversion of low to normal (low HDL-C at 2009 and normal HDL-C at 2013 and $\Delta\text{HDL-C} < 15\text{mg/dL}$)
- ◆ Conversion of low to increase (low HDL-C at 2009 and $\Delta\text{HDL-C} \geq 15\text{mg/dL}$)

Reference group: persistent normal HDL-C

(Results)

Baseline demographic and laboratory findings (p4)

In 2009, 4.614 million underwent both national health examination and gastric cancer screening. After the exclusion of subjects with any cancer diagnosed within 1 year, and those who died within 1 year, 4.535 million persons were enrolled at baseline. After excluding non-participants for the national general health examination in 2013, 2.869 million individuals were eligible. After further excluding absence of HDL-C values at follow-up, 2,865,902 persons are eligible [model III]. After further excluding any cancer up to 2013, 2,797,873 individuals (1,333,015 men [47.6%]; mean age of 54.0 years) were eligible [model I, main model] and followed up until 2017 (Fig. 1A).

Cancer risk by further categorization of HDL-C change (p 7)

We assessed individual cancer risk by HDL-C changes using ATP III and absolute HDL-C change (Δ HDL-C) (Fig 1C and Extended Data Table 2). Among baseline normal group, except for liver cancer, an increase over than 15mg/dL of HDL-C level from baseline had a null effect on the risk of gastrointestinal cancers compared with the persistently normal group. However, an increase > 15 mg/dL of HDL-C among baseline normal HDL-C group (aHR, 1.33; 95% CI, 1.19 - 1.48) as well as normal-to-low HDL-C group (aHR, 1.22; 95% CI, 1.11 - 1.37) had higher liver cancer risk compared to the persistently normal group (Fig 4). Persistent low group (low HDL-C at baseline and Δ HDL-C less than 15 mg/dL) increased the risk of overall, gastric, colorectal, and biliary cancer, whereas low-normal and low-increase group did not increase their risk. However, both persistent low (aHR 1.18; 95% CI 1.06-1.31) and low-to-increase (aHR 1.38; 95% CI 1.21-1.59) group increased liver cancer risk (Fig 4). In model III, the effect of HDL-C change on individual cancer risk was similar in main model (Supplementary Table 6)

(Discussion) (p 12)

The favorable effect of HDL-C on cancer appears to be related to its anti-inflammatory and anti-oxidative properties.^{12, 23} LDL-C in the tumor microenvironment can promote pro-oxidant and pro-inflammatory activities, whereas cholesterol and its metabolites can be removed from cancer cells by HDL-C.²⁴ HDL-C associated apolipoproteins and ATP-binding cassette transporters might have anti-tumorigenic effects.^{25, 26} HDL-C also has anti-apoptotic effects.²⁷ This anti-apoptotic effect of HDL-C may partially explain the increased risk of liver cancer in

the change from the normal to the high HDL-C group. Liver is a complex functional organ related to lipid metabolism and is one of organs of estrogen action.²⁸ Functional investigation for the link of HDL-C and liver cancer may clarify our epidemiologic findings. However, a previous study suggests that high cholesterol consumption by malignant cells to promote tumor growth might deplete plasma cholesterol and HDL-C levels.²⁹ Therefore, low HDL-C might be the epiphenomenon of cancer-related inflammation.^{30, 31} To avoid this issue, we enrolled a cancer-screened population and excluded patients diagnosed with any cancer before baseline and at baseline and excluded persons who diagnosed with 1 years after baseline enrollment (Model III). And we further excluded those who were diagnosed with any cancer until the second measurement of HDL-C (main model, model I). In addition, we used sensitivity analysis to evaluate reverse causation after excluding individuals who were diagnosed with any cancer until 1 years after the second measurement of HDL-C.

(Supplementary Table 6) Cancer risk by HDL-C change (Categorization B, model III)

			Model III	
HDL change	Total number (n=2865902)	Any cancer (n=176048)	HR (95% CI)	P-value
Low-Increase	130029	8065	1.04 (1.01-1.06)	0.003
Low-Low	338415	22175	1.10 (1.09-1.12)	<.0001
Low-Normal	194224	11848	1.02 (1.00-1.04)	0.017
Normal-Increase	185355	10992	1.00 (0.98-1.02)	0.800
Normal-Low	339878	23610	1.18 (1.16-1.19)	<.0001
Normal-Normal	1678001	99358	1	
	Total number (n=2865902)	C16 (n=25224)	HR (95% CI)	P-value
Low-Increase	130029	1136	0.97 (0.91-1.03)	0.331
Low-Low	338415	2714	0.90 (0.86-0.94)	<.0001
Low-Normal	194224	1627	0.94 (0.89-0.98)	0.010
Normal-Increase	185355	1772	1.07 (1.02-1.12)	0.008
Normal-Low	339878	3025	1.01 (0.97-1.05)	0.802
Normal-Normal	1678001	14950	1	
	Total number (n=2865902)	C25 (n=4739)	HR (95% CI)	P-value
Low-Increase	130029	214	1.05 (0.91-1.21)	0.489

Low-Low	338415	655	1.25 (1.14-1.36)	<.0001
Low-Normal	194224	338	1.12 (1.00-1.25)	0.054
Normal-Increase	185355	274	0.95 (0.84-1.08)	0.446
Normal-Low	339878	667	1.28 (1.17-1.39)	<.0001
Normal-Normal	1678001	2591	1	
	Total number (n=2865902)	C23 (n=1759)	HR (95% CI)	P-value
Low-Increase	130029	85	1.12 (0.90-1.40)	0.307
Low-Low	338415	236	1.21 (1.05-1.39)	0.009
Low-Normal	194224	105	0.93 (0.76-1.14)	0.507
Normal-Increase	185355	102	0.95 (0.78-1.17)	0.646
Normal-Low	339878	267	1.37 (1.20-1.57)	<.0001
Normal-Normal	1678001	964	1	
	Total number (n=2865902)	C24 (n=2283)	HR (95% CI)	P-value
Low-Increase	130029	131	1.32 (1.10-1.58)	0.003
Low-Low	338415	307	1.20 (1.06-1.36)	0.005
Low-Normal	194224	145	0.98 (0.83-1.17)	0.852
Normal-Increase	185355	127	0.91 (0.76-1.09)	0.290
Normal-Low	339878	311	1.22 (1.08-1.38)	0.002
Normal-Normal	1678001	1262	1	
	Total number (n=2865902)	CRC (n=21476)	HR (95% CI)	P-value
Low-Increase	130029	906	0.93 (0.87-1.00)	0.045
Low-Low	338415	2584	1.03 (0.99-1.07)	0.174
Low-Normal	194224	1372	0.95 (0.90-1.00)	0.070
Normal-Increase	185355	1293	0.94 (0.89-0.99)	0.031
Normal-Low	339878	2893	1.16 (1.11-1.20)	<.0001
Normal-Normal	1678001	12428	1	

CI, confidence interval; HDL-C, high density lipoprotein cholesterol; HR, hazard ratio. Low HDL-C refers to <40mg/dL in men and <50mg/dL in women. Normal HDL-C means \geq 40mg/dL in men and \geq 50mg/dL in women.

Increase means Δ HDL-C [(HDL-C at follow-up) - (HDL-C baseline)] \geq 15mg/dL. Detail definition was provided in Fig 1C and Supplementary material p5.

(Extended Table 2) Cancer risk by HDL-C change (Categorization B, Main model)

	Main model					
HDL change	Total number (n=2797873)	Any cancer (n=108019)	HR (95% CI)	P-value	aHR (95% CI)*	P-value
Low-Increase	126950	4986	1.02 (0.99-1.05)	0.12	1.02 (0.99-1.06)	0.162
Low-Low	329514	13274	1.06 (1.04-1.08)	<.0001	1.05 (1.03-1.07)	<.0001
Low-Normal	189662	7286	1.01 (0.99-1.03)	0.453	1.02 (0.99-1.04)	0.231
Normal-Increase	181314	6951	1.01 (0.99-1.04)	0.295	1.02 (0.99-1.04)	0.274
Normal-Low	330092	13824	1.11 (1.09-1.13)	<.0001	1.07 (1.05-1.10)	<.0001
Normal-Normal	1640341	61698	1		1	
	Total number (n=2797873)	C16 (n=14649)	HR (95% CI)	P-value	aHR (95% CI)*	P-value
Low-Increase	126950	627	0.91 (0.84-0.99)	0.02	1.07 (0.98-1.16)	0.162
Low-Low	329514	1629	0.92 (0.87-0.97)	0.001	1.07 (1.01-1.14)	0.031
Low-Normal	189662	969	0.95 (0.89-1.01)	0.115	1.04 (0.96-1.12)	0.339
Normal-Increase	181314	894	0.92 (0.86-0.99)	0.019	0.94 (0.87-1.01)	0.105
Normal-Low	330092	1800	1.02 (0.97-1.07)	0.431	1.08 (1.02-1.14)	0.007
Normal-Normal	1640341	8730	1		1	
	Total number (n=2797873)	C22 (n=6574)	HR (95% CI)	P-value	aHR (95% CI)†	P-value
Low-Increase	126950	331	1.10 (0.99-1.24)	0.086	1.38 (1.21-1.59)	<.0001
Low-Low	329514	689	0.89 (0.82-0.97)	0.006	1.18 (1.06-1.31)	0.0021
Low-Normal	189662	396	0.89 (0.80-0.99)	0.028	1.03 (0.90-1.17)	0.6771
Normal-Increase	181314	525	1.24 (1.13-1.36)	<.0001	1.33 (1.19-1.48)	<.0001
Normal-Low	330092	827	1.08 (1.00-1.16)	0.054	1.22 (1.11-1.37)	<.0001
Normal-Normal	1640341	3806	1		1	
	Total number (n=2797873)	C25 (n=3658)	HR (95% CI)	P-value	aHR (95% CI)*	P-value
Low-Increase	126950	164	1.02 (0.87-1.19)	0.829	0.86 (0.72-1.03)	0.096
Low-Low	329514	502	1.21 (1.10-1.33)	0	1.04 (0.93-1.16)	0.537
Low-Normal	189662	247	1.04 (0.91-1.18)	0.61	0.94 (0.81-1.08)	0.37
Normal-Increase	181314	206	0.91 (0.79-1.05)	0.191	0.94 (0.80-1.09)	0.41
Normal-Low	330092	500	1.21 (1.10-1.34)	0	1.09 (0.98-1.21)	0.112
Normal-Normal	1640341	2039	1		1	
	Total number (n=2797873)	C23 (n=1304)	HR (95% CI)	P-value	aHR (95% CI)*	P-value
Low-Increase	126950	65	1.16 (0.90-1.49)	0.255	1.05 (0.80-1.38)	0.746
Low-Low	329514	178	1.23 (1.05-1.45)	0.013	1.02 (0.85-1.24)	0.805
Low-Normal	189662	83	1.00 (0.80-1.25)	0.995	0.95 (0.74-1.22)	0.685
Normal-Increase	181314	77	0.98 (0.77-1.23)	0.837	1.04 (0.81-1.34)	0.737
Normal-Low	330092	191	1.33 (1.14-1.56)	0	1.20 (1.01-1.43)	0.04
Normal-Normal	1640341	710	1		1	
	Total number (n=2797873)	C24 (n=1763)	HR (95% CI)	P-value	aHR (95% CI)*	P-value
Low-Increase	126950	88	1.11 (0.89-1.38)	0.368	1.07 (0.84-1.35)	0.607
Low-Low	329514	237	1.16 (1.01-1.33)	0.043	1.12 (0.96-1.32)	0.162
Low-Normal	189662	118	1.00 (0.83-1.21)	0.983	0.99 (0.80-1.22)	0.893

Normal-Increase	181314	100	0.89 (0.73-1.1)	0.288	0.81 (0.64-1.02)	0.071
Normal-Low	330092	215	1.06 (0.91-1.23)	0.454	0.97 (0.82-1.14)	0.689
Normal-Normal	1640341	1005	1		1	
	Total number (n=2797873)	CRC (n=12435)	HR (95% CI)	P-value	aHR (95% CI)*	P-value
Low-Increase	126950	514	0.91 (0.83-1)	0.039	0.94 (0.85-1.04)	0.222
Low-Low	329514	1551	1.07 (1.01-1.13)	0.024	1.09 (1.02-1.16)	0.01
Low-Normal	189662	804	0.96 (0.89-1.03)	0.269	0.98 (0.90-1.06)	0.541
Normal-Increase	181314	780	0.98 (0.91-1.05)	0.573	0.97 (0.89-1.05)	0.425
Normal-Low	330092	1613	1.11 (1.06-1.18)	<.0001	1.09 (1.02-1.15)	0.006
Normal-Normal	1640341	7173	1		1	

*Adjusted for age, sex, economic status, body mass index, hypertension, diabetes, cerebrovascular disease, heart disease, smoking status, drinking status, physical activity, and use of lipid lowering drug.

†Adjusted for age, sex, economic status, body mass index, hypertension, diabetes, cerebrovascular disease, heart disease, smoking status, drinking status, physical activity, use of lipid lowering drug, liver factors (chronic liver disease, chronic hepatitis B, chronic hepatitis C, and liver cirrhosis) and triglyceride.

CI, confidence interval; HDL-C, high density lipoprotein cholesterol; HR, hazard ratio. Low HDL-C refers to <40mg/dL in men and <50mg/dL in women. Normal HDL-C means ≥40mg/dL in men and ≥50mg/dL in women.

(Figure 1)

(C) HDL-C change categorization : 4 groups by ATP III (main categorization)

		HDL-C at 2013	
		Low	Normal
HDL-C at 2009	Low	Low → Low	Low → Normal
	Normal	Normal → Low	Normal → Normal (ref)

HDL-C change categorization : 6 groups

		HDL-C at 2013		ΔHDL-C≥15mg/dL
		Low	Normal	
HDL-C at 2009	Low	Low → Low and ΔHDL-C<15 mg/dL	Low → Normal and ΔHDL-C<15 mg/dL	Low → Increase ΔHDL-C≥15mg/dL
	Normal	Normal → Low	Normal → Normal and ΔHDL-C<15 mg/dL (ref)	Normal → Increase ΔHDL-C≥15mg/dL

Absolute HDL change [ΔHDL-C = (HDL-C at 2013) - (HDL-C at 2009)], mg/dL

Decrease		Stable		Increase	
< -10	-10 ~ -5	-5 ~ 5 (ref)	5 ~ 15	15 ~ 25	≥25

(Figure 4) Categorization B, main model

2. The most important limitation is the lack of mutual adjustment for other lipoproteins. As the different cholesterol particles are correlated, it is not clear if the results are due to HDL or other lipoproteins.

Answer> Thank you for the kind comment.

Total cholesterol was not associated with GI cancer risk in this cohort. Although Low-density lipoprotein cholesterol (LDL-C) and triglycerides (TG) are affected by long-term exercise, diet, and obesity, LDL-C is also strongly affected by short-term lipid-lowering drugs (Stone et al, 2014) and TG are strongly influenced by recent diet and alcohol consumption. On the other hand, HDL-C is influenced by regular exercise, obesity, and diet pattern rather than short-term diet, exercise, and lipid lowering drugs. This relative stability of HDL-C can be used as a marker for cancer risk assessment, so we focused on HDL-C. Actually our previous studies showed no association between gastric cancer and other lipids (total cholesterol, triglyceride, and LDL) (Nam et al. UEG journal 2019, GIE 2019). However, triglyceride was significant associated with liver cancer risk in unadjusted analysis. Therefore liver cancer analysis was additionally adjusted for liver related factors including viral hepatitis, liver cirrhosis, and TG.

(ref) Stone NJ, Robinson JG, Lichtenstein AH, et al. 2013 ACC/AHA guideline on the treatment of blood cholesterol to reduce atherosclerotic cardiovascular risk in adults: a report of the American College of Cardiology/American Heart Association Task Force on Practice Guidelines. *Circulation*. 2014;129(25 Suppl 2):S1-45.

(ref) Nam SY, Park BJ, Nam JH et al. Association of current *Helicobacter pylori* infection and metabolic factors with gastric cancer in 35,519 subjects: A cross-sectional study. *United European Gastroenterol J*. 2019;7(2):287-296.

(ref) Nam SY, Park BJ, Nam JH et al. Effect of *Helicobacter pylori* eradication and high-density lipoprotein on the risk of de novo gastric cancer development. *Gastrointest Endosc*. 2019 Sep;90(3):448-456.e1.

We added related paragraph in Discussion part. (p12-13)

Associations between gastrointestinal cancer risk and other lipids such as total cholesterol, LDL-C, triglyceride have also been studied.^{5,6,8,10} A cohort study from Finland suggested that HDL-C was inversely associated with the risk of several site cancers but total cholesterol had no association with cancer risk during long term follow-up.⁵ A US women cohort study showed that HDL-C was inversely associated with colorectal and lung cancer but LDL-C was not significantly associated with risk of total cancer or any site-specific cancers.⁹ LDL and triglycerides are affected not only by long-term exercise, diet, and obesity, but also by short-term lipid-lowering drugs in case of LDL-C,³² and by recent diet and alcohol consumption in case of triglycerides. On the other hand, HDL-C is influenced by long term exercise and diet pattern rather than short-term diet or lipid lowering drugs. This relative stability of HDL-C may be used as a marker for cancer risk assessment, so we focused on HDL-C. Actually our previous studies showed no association between gastric cancer and other lipids (total cholesterol, triglyceride, and LDL).^{16, 33} However, triglyceride was significant associated with liver cancer risk in the current study. Therefore liver cancer analysis was additionally adjusted for triglyceride.

We added the comments in statistical analysis (p26).

We performed a multivariate analysis adjusted for significant confounders, such as age, sex,

BMI, smoking status, alcohol intake, physical activity, economic status, use of lipid lowering drugs, hypertension, diabetes, cerebrovascular disease, heart disease, and physical activity. **Triglyceride was significantly associated with liver cancer risk in unadjusted analysis.** Therefore, liver cancer analysis was additionally adjusted for liver factors, such as liver disease, chronic hepatitis B, chronic hepatitis C, liver cirrhosis, and triglyceride level. A directed acyclic graph provides an assumed causal framework for covariate adjustments (**Supplementary Fig. 1**).

We provided further adjusted analysis for liver cancer (Table 2).

Liver cancer[†]	Low → Low	705	0.88 (0.81-0.95)	0.001	1.15 (1.04-1.28)	0.007
	Low → Normal	711	0.95 (0.87-1.03)	0.18	1.12 (1.01-1.24)	0.03
	Normal → Low	827	1.05 (0.98-1.13)	0.19	1.18 (1.07-1.29)	<0.001
	Normal → Normal	4,331	1		1	

[†]Adjusted for age, sex, economic status, body mass index, hypertension, diabetes, cerebrovascular disease, heart disease, smoking status, drinking status, physical activity, use of lipid lowering drug, liver factors (chronic liver disease, chronic hepatitis B, chronic hepatitis C, and liver cirrhosis) and triglyceride.

3. Introduction, the authors report briefly on the association of HDL with CVD, but this association has been challenged on whether it is causal or not, and other lipoproteins seem more important.

Answer> Thank you for the kind comment.

As reviewer's comment, there is a limited evidence on whether low level of HDL-C is causal or not in the risk of cardiovascular disease. Therefore, we described as just association not causal relation. High TG and LDL-C are also associated with CVD. Our study focused on GI cancer not on CVD. **We revised the Introduction, as reviewer's recommendation (p 3).**

High-density lipoprotein cholesterol (HDL-C) is inversely associated with cardiovascular diseases or mortality^{1, 2} and non-HDL lipid such as triglyceride and low density lipoprotein cholesterol (LDL-C) is positively associated with them.¹

4. Methods, it is not clear why the study started in 2009 and not earlier.

Answer> Thank you for the important comment. HDL-C was measured from 2009 in National Health Insurance System. The year of 2009 is the earliest time point to conduct studies using HDL-C. **We described this in Method (p25).**

From 2009 to 2017, National General Health Examination provided measurement of HDL-C to all Korean persons aged 40 years or older every two years.

5. Methods, the authors reported "Patients who had been diagnosed with cancer before 2009 and within 1 year of baseline enrollment, and those who died within 1 year of enrollment were excluded" It is unclear how prevalent cancers were assessed and how complete this assessment is.

Answer> Thank you for the important comment. We are sorry for the unclear description.

NHISS provided data after excluding C-codes from 2004-2007 and we further excluded C-codes within 12 months from index month (the month of health examination). And we excluded the persons who died within 1 year of enrollment were excluded. Death data from the National Statistical Office were also provided by the National Health Insurance Service.

Nearly complete sensing of new cancer is possible through extracting C code from NHISS. In order to receive a special exemption for cancer, the cancer code should be registered with diagnostic evidence of cancer such as pathologic results at hospital. If any cancer code is registered, the patients pay just 5% for almost medical services. Therefore, cancer (C) code extracted from T20 is highly reliable.

We clarified the Method (P 25).

NHISS provided raw data after excluding persons with gastrointestinal cancer from 2004 to 2007. Subjects who both underwent NGHE (4.983 million) and cancer screening (4.75 million) in 2009 and NGHE in 2013 were included in this study. Among subjects who underwent both NGHE and National Cancer Screening from January to December 2009 (4.614million), we excluded persons with any C-codes within 12 months from index month (the month of health

examination) (n=65,791) and the persons who died within 1 year of enrollment (n=14,863) (**Fig. 1A**). Death data from the National Statistical Office were also provided by the National Health Insurance Service. Nearly complete sensing of new cancer is possible through extracting C code form NHISS. In order to receive a special exemption for cancer, the cancer code should be registered with diagnostic evidence of cancer such as pathologic results at hospital. If any cancer code is registered, the patients pay just 5% for almost medical services. Therefore, cancer (C) code extracted from NHISS is highly reliable. Individuals who did not undergo NGHE in 2013 (n= 1,665,999) or absence of HDL-C values in 2013 were excluded [**model III; n=2,865,902**]. And persons who had been diagnosed with cancer (n=68,029) up to 2013 were further excluded [**model I, main model; n=2,797,873**] (**Fig 1B**).

Reviewer #2 (Remarks to the Author):

Re: Impact of Longitudinal Change of High-density Lipoprotein Cholesterol on Gastrointestinal Cancer Risk in a Large Cohort

The authors have explored the association between changes in HDL-C and risk of gastrointestinal cancer, using a cohort of 2.8 million cancer-free individuals followed for 8 years. This is a great resource, and this does add novel findings on the impact of HDL-c on cancer risk. For the most part the paper is well written. I do recommend two additional analyses for improvement of this study: (1) show results by both sex and smoking status (i.e., female smokers, female nonsmokers, male smokers etc.) and (2) Incorporate baseline HDL-c level in the analysis of HDL-c continuous change. I recommend supplying a supplementary table that shows how excluded participants (those without a second measurement) differed from the individuals included in the paper. The discussion could be improved upon by focusing less on repeating direct results and more on interpretation of findings. Specific comments are provided for each section below:

Answer> We really thank you so much for the valuable comments. We revised the manuscript and provided point-by-point responses to each comment.

Abstract

1. Please include a sentence about the methods (time to event analysis, HR calculation, adjustment factors). Note that you have separated smokers and non-smokers as well, so that the conclusions follow.

Answer> Thank you for the kind comment. We skipped the above contents due to Abstract word limitation. **We added brief comments for methods and removed redundant contents in the abstract. We further described subgroup analysis by smoking status and removed absolute HDL-C change (p2).**

The impact of high-density lipoprotein cholesterol (HDL-C) change on cancer has not been reported. Here, we investigated the impact of HDL-C level change on gastrointestinal cancer risk in a large population-based cohort. Cancer-free individuals, who underwent health examinations including HDL-C in 2009 and 2013 were followed-up through 2017. HDL-C changes were classified into persistent normal (normal-to-normal), normal-to-low, low-to-normal, and persistent low (low-to-low) HDL-C group. **Cancer risk according to HDL-C change was measured with hazard ratios (HRs) and 95% confidence intervals (CIs) using Cox proportional regression analysis.** Among 2.8 million persons, 108,102 patients developed any type of cancer including 14649 gastric, 12435 colorectal, 6574 liver, 3662 pancreatic, 1331 gallbladder, and 1765 biliary cancers. The persistent low HDL-C group had higher risk for overall (**aHR 1.06; 95% CI 1.03-1.07**), gastric (**aHR 1.07; 95% CI 1.01-1.14**), liver (**aHR 1.15; 95% CI 1.04-1.28**), and colorectal cancer (**aHR 1.08; 95% CI 1.02-1.15**) comparing to persistent normal HDL-C group. HDL-C change from normal to low level increased the risk for overall (**aHR 1.07; 95% CI 1.05-1.09**), gastric (**aHR 1.09; 95% CI 1.03-1.15**), colorectal (**aHR 1.09; 95% CI 1.03-1.16**), liver (**aHR 1.18; 95% CI 1.07-1.29**) and gallbladder cancers (**aHR 1.19; 95% CI 1.01-1.42**). **The persistent low and normal-to-low HDL-C groups showed increased gastric and liver cancer risk only in ever-smokers. The persistently low HDL-C group had increased colorectal cancer risk in both never and ever smokers, whereas the normal-to-low HDL-C group increased colorectal cancer risk only in ever-smokers.** In conclusion, persistently low HDL-C and HDL-C change from normal to low levels increased gastrointestinal cancer risk with discrepancies by smoking status.

Introduction

1. In the second paragraph it would be worthwhile to include a reason for investigating

the change in HDL-C levels- i.e., what evidence is there that suggests the change over time may be important for risk? I believe more can be added to the introduction to make a case for this study.

Answer> Thank you for the important comment. **We added reason for investigating the change in HDL-C levels in the Introduction (p2).**

As mentioned above, **baseline** HDL-C levels are inversely associated with the risk of several cancers **in cohort studies** but have no significant association with some cancers. **The normal HDL-C at the baseline is often maintained, but the HDL-C may decrease significantly in some persons. Low HDL-C will also be maintained in many cases, but may be improved to normal HDL-C in some people. The effects of these HDL-C changes on cancer development have not been reported.** We investigated the impact of longitudinal changes in HDL-C level on gastrointestinal cancer risk using a large database from the Korean National Health Insurance Service System (NHISS).

2. The last few sentences of the introduction are methods focused and could be abbreviated.

Answer> Thank you for the kind comment. As journal policy, Method is placed in the last part. Therefore, we have briefly summarized our study plan in the Introduction part for easy understanding to readers. **We revised the Introduction (p2).**

We investigated the impact of persistent exposure to low HDL-C level and longitudinal changes in HDL-C level (change from normal to low HDL-C level and from low to normal HDL-C level) on gastrointestinal cancer risk using a large database from the Korean National Health Insurance Service System (NHISS). Several cancer studies using the Korean NHISS have been published.^{13, 14} The definition of normal and low HLD-C was defined according to the Adult Treatment Panel III (ATP III).¹⁵ Furthermore, we conducted interaction analysis between HDL-C change and important cofactors in cancer risk and performed sub-group analyses to determine their associations by sex and smoking status. We also investigated the effect of absolute HDL-C change (decrease, stable, and increase) on cancer risk.

→ We investigated the impact of longitudinal changes in HDL-C level on gastrointestinal

cancer risk using a large database from the Korean National Health Insurance Service System (NHISS). Several cancer studies using the Korean NHISS have been published.^{13, 14} The definition of normal and low HDL-C was defined according to the Adult Treatment Panel III (ATP III).¹⁵ Furthermore, we conducted interaction analysis between HDL-C change and important cofactors in cancer risk and performed sub-group analyses to determine their associations by sex and smoking status.

Results

1. Subgroup analysis among baseline normal HDL-C groups is hard to follow. Why was an increase in HDL-C tested for individuals who were in the normal-to-low group? None of these patients should have an increase in HDL-c correct? Please clarify this in the results and in the methods sections.

Answer> Thank you for the kind comment. We would like to know which further increase of HDL-C among baseline normal HDL-C can change the risk of gastrointestinal cancer?

We revised Results, Method and Supplementary Material to clarify this issue.

Results

Cancer risk by further categorization of HDL-C change (p4)

We assessed gastrointestinal cancer risk according to HDL-C change category using ATP III and absolute HDL-C change (Δ HDL-C) (**Fig 1C and Extended Data Table 2**). Among **baseline normal group**, except for liver cancer, an increase over than 15mg/dL of HDL-C level from baseline had a null effect on the risk of gastrointestinal cancers compared with the persistently normal group. However, an increase > 15 mg/dL of HDL-C among baseline normal HDL-C group (aHR, 1.33; 95% CI, 1.19 - 1.48) as well as normal-to-low HDL-C group (aHR, 1.22; 95% CI, 1.11 - 1.21) had a higher liver cancer risk compared to the persistently normal group (**Fig 4**). **Persistent low HDL-C group (low HDL-C at baseline and Δ HDL-C < 15 mg/dL) increased the risk of overall, gastric, colorectal, and biliary cancer, whereas low-normal and low-increase HDL-C group did not increase their risk. However, both persistent low HDL-C (aHR 1.18; 95% CI 1.06-1.31) and low-to-increase HDL-C (aHR1.38; 95% CI 1.21-1.59) increased liver cancer risk.**

Methods (p27)

We further categorized HDL-C change using ATP III and absolute HDL-C change [Δ HDL-C = (HDL-C at follow-up) - (HDL-C baseline)] to investigate which further increase of HDL-C among baseline normal HDL-C can change the risk of gastrointestinal cancer (**Fig. 1C and Supplementary material p5**). Persistent normal group was further classified into persistent normal and normal-increase (baseline normal and increase ≥ 15 mg/dL at follow-up from baseline) groups. Baseline low group was classified as persistent low, low-to-normal, and low-to-increase.

Supplementary p 5.

Change of HDL-C was classified into 6 groups using APTII and Δ HDL-C

- ◆ Persistent normal HDL-C (normal HDL-C at 2009 and normal HDL-C at 2013 and Δ HDL-C <15mg/dL)
- ◆ Conversion of normal to low HDL-C (normal HDL-C at 2009 and low HDL-C at 2013)
- ◆ Normal to increase HDL-C (normal HDL-C at 2009 and Δ HDL-C ≥ 15 mg/dL)
- ◆ Persistent low (low HDL-C at 2009 and low HDL-C at 2013 and Δ HDL-C <15mg/dL)
- ◆ Conversion of low to normal (low HDL-C at 2009 and normal HDL-C at 2013 and Δ HDL-C <15mg/dL)
- ◆ Conversion of low to increase (low HDL-C at 2009 and Δ HDL-C ≥ 15 mg/dL)

Reference group: persistent normal HDL-C

2. Subgroup analysis by sex: Please specify what change you are referring to in the following sentences: “A change in the normal to low HDL-C group was associated with an increased risk of liver cancer in men (aHR 1.20; 95% CI 1.06-1.36) and a change in the low to normal HDL-C group was associated with a reduction in women’s liver cancer risk (aHR 1.32 0.83; 95% CI 0.69-0.99).” “A change in the normal to low HDL-C group was associated with an increased risk of pancreatic cancer in women (aHR 1.19; 95% CI

1.02-1.39) and had a borderline association with women's gallbladder cancer risk (aHR 1.25; 95% CI 0.97-1.62)".

a. Is it meant to state that "Those in the normal- to-low HDL-C group had an increased risk of liver cancer....". If that is the case, please reword; as it is stated it seems there is an additional change involved here.

Answer> Thank you for the kind comment. "A change in the normal to low HDL-C group" means "normal HDL-C at baseline and low HDL-C at follow up". We used both "normal-to-low" and "change in the normal to low HDL-C group". This may make a confusion. **We clarified HDL-C change group (P6).**

Normal-to-low HDL-C group was associated with an increased risk of liver cancer in men (aHR 1.20; 95% CI 1.06-1.36), but normal-to-low HDL-C group was associated with a reduction in women's liver cancer risk (aHR 0.83; 95% CI 0.69-0.99) (Fig 3B). The persistently low HDL-C group increased women's colorectal cancer risk (aHR 1.16; 95% CI 1.06-1.27), whereas the impact of HDL-C changes was less remarkable in men (Fig 3C). Normal-to-low HDL-C group was associated with an increased risk of pancreatic cancer in women (aHR 1.19; 95% CI 1.02-1.39) and had a borderline association with women's gallbladder cancer risk (aHR 1.25; 95% CI 0.97-1.62).

3. Effect of absolute HDL-C change on cancer risk: The following statement is unclear, please reword; I believe instead of increment, you meant "increase":

a. "An increment of HDL-C at follow-up was associated the reduced risk of gastric cancer (HR, 0.96 for Δ HDL C 5-15 mg/dL and 0.94 for Δ HDL-C 15-25 mg/dL) (Fig. 4B)."

Answer> Thank you for the kind comment. **We clarified the sentence (P8).**

An increase of HDL-C at follow-up was associated the reduced risk of gastric cancer (HR, 0.96 for Δ HDL-C 5-15 mg/dL and 0.94 for Δ HDL-C 15-25 mg/dL) (Fig. 5B).

4. In Extended Data Table 2, "HDL-G" is written – should be "HDL-C"

Answer> Thank you for the kind comment.

HDL_G means HDL-C change group. **We commented as footnote below the Table.**

DF, degree of freedom; HDL_G, HDL-C change group (4 groups); SMK, smoking status (never, past, current smokers).

5. Given the interactions of both sex and smoking seen in this study, I recommend reporting results broken down by sex- and smoking status together.

Answer> Thank you for the important comment.

The portion of current smokers is 34% in men and 2.2% in women. Event numbers in women smokers were very low. Therefore we did not conduct subgroup analysis by sex and smoking together. We also considered subgroup analysis by menopausal status. Interaction analysis showed no significance and the event number in premenopausal women in some group was too small. Therefore, we did not conduct subgroup analysis. We adjusted for smoking in sub-group analysis by sex and adjusted for sex in sub-group analysis by smoking. To conduct further subgroup analysis by sex- and smoking status together (men-smoker, men-never smoker, women-smoker, women-never smoker), we need raw data set. However, the raw data set is expired. It will take 1 year to get the data set again from NHIS.

We stated this analysis in Limitation (p 14).

Third, subgroup analysis by sex- and smoking status together (men-smoker, men-never smoker, women-smoker, women-never smoker) may provide more precise association between HDL-C change and cancer risk. But, the portion of current smokers is 34% in men and 2.2% in women. Event numbers in women smokers were very low. Therefore, we did not conduct subgroup analysis by sex and smoking together. We adjusted for smoking in sub-group analysis by sex and adjusted for sex in sub-group analysis by smoking.

6. For figure 4, where change in HDL is measured continuously, it would be advantageous to break this down further based on initial HDL-C group. i.e., report the results for change seen among individuals with (a) normal HDL-c at baseline (b) low HDL-c at baseline, and (c) high HDL-c at baseline.

Answer> Thank you for the important comment.

We used several categorization to analysis the effect of HDL-C change on cancer risk.

Fig 4 (revised Fig 5) categorization is the initial categorization for analysis. However, we think that simple change between baseline and follow-up is not enough to investigate their association. For examples, stable population includes stable low HDL-C and stable normal HDL-C.

Therefore, we further categorized HDL-C change.

Categorization A: Δ HDL-C (revised Fig 5)

Categorization B: using APT III and Δ HDL-C (revised Fig 4)

		2013 HDL-C		
		Low	Normal	Increase ≥ 15
2009 HDL-C	Low	338415 (Low \rightarrow low and Δ HDL-C $<$ 15 mg/dL)	194224 (low \rightarrow normal and Δ HDL-C $<$ 15 mg/dL)	130029 (Δ HDL-C \geq 15mg/dL)
	Normal	339878	1678001 (reference)	185355 (Δ HDL-C \geq 15mg/dL)

HDL-C change categorization : 6 groups

		HDL-C at 2013		
		Low	Normal	Δ HDL-C \geq 15mg/dL
HDL-C at 2009	Low	Low \rightarrow Low and Δ HDL-C $<$ 15 mg/dL	Low \rightarrow Normal and Δ HDL-C $<$ 15 mg/dL	Low \rightarrow Increase Δ HDL-C \geq 15mg/dL
	Normal	Normal \rightarrow Low	Normal \rightarrow Normal and Δ HDL-C $<$ 15 mg/dL	Normal \rightarrow Increase Δ HDL-C \geq 15mg/dL

Categorization C (main analysis)

HDL-C change : 4 groups by ATP III

		HDL-C at 2013	
		Low	Normal
HDL-C at 2009	Low	Low \rightarrow Low	Low \rightarrow Normal
	Normal	Normal \rightarrow Low	Normal \rightarrow Normal (ref)

As reviewer's comment, the effect of HDL change on cancer risk among individuals with (a) normal HDL-c at baseline (b) low HDL-c at baseline, and (c) high HDL-c at baseline can provide more information. We should conduct further analysis. However, raw data was expired and it takes 1 year to get the data set again from NHISS. We are sorry for this limitation. Instead of this, **we provide the categorization B [(baseline normal-to-low, normal, increase) and (baseline low-to-low, normal, increase)] (Fig 1C, revised Fig 4).**

Fig 1

Fig 4

Discussion

1. The discussion is a bit choppy and often repeats results without adding relevant context, e.g.: **“The hazardous effect of persistent low and change from normal to low HDL-C on liver cancer was significant only in smokers.”** – This statement does not explain the role of smoking in the association between HDL-c change and cancer risk and simply repeats results. It is not linked with the discussion on smoking below (lines 209-217), which would improve the flow.

Answer> Thank you for the important comment.

We moved the sentence “The hazardous effect of persistent low and change from normal to low HDL-C on liver cancer was significant only in smokers.” to smoking paragraph and more discussed interaction of smoking and HDL-C change in the cancer risk.

We revised the Discussion (p11).

Our main results were robust to a range of sensitivity analyses for the impact of exposure duration on most cancers, except liver cancers. The effects of HDL-C change on liver cancer were less prominent in model II. Subgroup analysis by sex showed that the effect of HDL-C change on overall cancer risk was similar in men and women. However, the effect of HDL-C change on colorectal and pancreatic cancer risk was remarkable in women.

Interestingly, normal-to-low HDL-C increased the risk of men’s liver cancer and low-to-normal HDL-C reduced the risk of women’s liver cancer. In the subgroup analysis by smoking status, the hazardous effect of persistently low HDL-C and change from normal to low HDL-C level on gastric and liver cancers was significant only in ever-smokers. **The hazardous effect of low-to-normal HDL-C on liver cancer was also significant only in ever-smokers. Even if persistent low HDL-C increased CRC risk in both never and ever smokers,** the harmful effect of HDL-C change from normal to low levels on colorectal cancer risk was observed only in ever-smokers. These results suggest the importance of maintaining normal HDL-C, particularly in ever-smokers. The mechanism of this discrepant effect of HDL-C change according to sex and smoking status in several cancers is unknown. A plausible explanation is that the immunologic and inflammatory response to smoking further promotes

a low HDL-C level–related cancer risk in current smokers. The related mechanisms need to be investigated in the future. Previous epidemiologic studies have reported sex-discrepant effects of BMI on cancer risk.²⁰ The interaction between BMI and smoking status on cancer risk has also been previously reported.^{20,21} **In a previous small study (259 cases of lung cancer in 14,547 members of atherosclerotic risk) using binary HDL-C, low HDL-C at baseline on lung cancer was associated with lung cancer only in past smokers (HR 1.77).²²**

2. I believe the flow and the quality of the discussion can be improved by focusing more on potential explanations for findings and previous literature and by limiting repetition of results. For example, focus on what results agree or disagree with previous literature, without restating all the results. Lines 224-231 add interesting insight into mechanisms and could be expanded upon.

Answer> Thank you for the important comment. **We have thoroughly revised the discussion (p9-10, 12).**

Persistent exposure to low HDL-C and change from normal to low HDL-C increased the risk of overall, gastric, and colorectal cancer (5-9%). These effect was more prominent in ever smokers (increase of risk with 6-18%). The impact of longitudinal changes in HDL-C on cancer risk has not been reported, even though an association between baseline HDL-C level and cancer risk has been reported (**Supplementary Table 8**). Low baseline HDL-C levels are positively associated with the overall cancer risk.^{5,6,9} A low baseline HDL-C levels increased the risk of gastric cancer in a cohort study adjusted for *H. pylori* and demographic factors¹⁶ and HDL-C was inversely associated with gastric cancer among postmenopausal women.¹⁷ A Finland study showed no significance between HDL-C and gastric cancer risk.⁵ HDL-C was not associated with colorectal cancer in a UK cohort study,⁸ whereas HDL-C was inversely associated with colorectal cancer in a US Women’s Health Study⁶ **and in European cohort study.¹⁸** **This non-constant association between baseline HDL-C and cancer risk may be related to size and portion of study population and study design. However, in the current study, normal HDL-C level at the baseline was maintained in many case (84%) at follow-up, and HDL-C decreased to low levels in small portion (16%). And the effect of these two groups on gastrointestinal cancer risk was far different. Interestingly, low HDL-C persisted just in 52% at follow-up, but HDL-C level increased to the normal range of HDL-C in 48%. It seems that people who have found out that they have low HDL-C through screening are**

making efforts to reach normal HDL-C. And the persistent low HDL-C increased the risk of gastrointestinal cancer, but low-to-normal group did not increase the cancer risk. Therefore, there are limitations to basal HDL-C-based cancer risk estimation performed in previous studies. Furthermore, our findings suggest that persons with low HDL-C at baseline need not be discouraged, and that improving the lipid profile with normal HDL-C can reduce cancer risk to a near extent to people with normal HDL-C at baseline.

The persistent exposure to low HDL-C (HR 1.15), change from normal to low HDL-C (aHR 1.18), and change from low to normal HDL-C (HR 1.12) had a higher liver cancer risk compared to the persistently normal HDL-C group. The finding that low-to-normal HDL-C level also increased the risk of cancer was observed only in liver cancer and was unexpected. Therefore, we further categorized HDL-C change group (Fig 4). A low-to-increase HDL-C change group (low HDL-C at baseline and Δ HDL-C over than 15 mg/dL) increased liver cancer risk, but low-to-normal HDL-C change group (low HDL-C at baseline and normal range of HDL-C at follow-up and Δ HDL-C less than 15 mg/dL) did not increased liver cancer risk. Furthermore, HDL-C level increase by > 15 mg/dL among the baseline normal HDL-C group (HR, 1.33) as well as the normal to low HDL-C group (HR, 1.22), increased liver cancer risk compared to the persistently normal group. Our findings suggest that extremely high HDL-C levels may increase the risk of liver cancer and emphasize that maintaining modest normal HDL-C levels is important to reduce the risk of liver cancer. The association between HDL-C level and liver cancer risk remains controversial. HDL-C levels were inversely associated with liver cancer risk in Finland⁵ and Swedish cohort studies,¹⁰ whereas a Danish cohort study showed no association between HDL-C and liver-biliary cancer.⁹ Our finding that maintenance of modest HDL-C level is important to minimize the liver cancer risk, may partly explain the controversial results of previous studies on the association between baseline HDL-C and liver cancer risk.

The favorable effect of HDL-C on cancer appears to be related to its anti-inflammatory and anti-oxidative properties.^{12, 23} LDL-C in the tumor microenvironment can promote pro-oxidant and pro-inflammatory activities, whereas cholesterol and its metabolites can be removed from cancer cells by HDL-C.²⁴ HDL-associated apolipoproteins and ATP-binding cassette transporters might have anti-tumorigenic effects.^{25,26} HDL-C also has anti-apoptotic effects.²⁷ This anti-apoptotic effect of HDL-C may partially explain the increased risk of liver

cancer in the change from the normal to the high HDL-C group. Liver is a complex functional organ related to lipid metabolism and is one of organs of estrogen action.²⁸ Functional investigation for the link of HDL-C and liver cancer may clarify our epidemiologic findings. However, a previous study suggests that high cholesterol consumption by malignant cells to promote tumor growth might deplete plasma cholesterol and HDL-C levels.²⁹ Therefore, low HDL-C might be the epiphenomenon of cancer-related inflammation.^{30, 31} To avoid this issue, we enrolled a cancer-screened population and excluded patients diagnosed with any cancer before baseline and at baseline and excluded persons who diagnosed with 1 years after baseline enrollment (Model III) or those who were diagnosed with any cancer until the second measurement of HDL-C (main model, model I). In addition, we used sensitivity analysis to evaluate reverse causation after excluding individuals who were diagnosed with any cancer until 1 years after the second measurement of HDL-C.

Methods

1. **Given the exclusion of individuals who did not have a second measurement (undergo an examination in 2013), I believe it needs to be clearly stated throughout the paper that this study only applies to individuals who have been followed (remained cancer-free) for at least four years.** This speaks to the generalizability of the findings, though it was not made clear throughout interpretation of results. **Further, was risk of cancer examined among those individuals who did not undergo a health examination in 2013?** It would be of interest to know **how these individuals compared with those who were included in the study (e.g., comorbidities, initial HDL-C levels, demographics, cancer outcomes) to understand better who is excluded. (also mentioned in my comments on “Results” above)**

Answer> Thank you for the kind comment.

We provided the number of each step after exclusion in Figure 1A and results to avoid duplication. HDL-C level at 2013 can be extracted from persons who underwent general health examination at 2013. Therefore, we excluded persons who did not undergo general health examination at 2013. **We provided baseline characteristics of excluded and included persons in Supplementary Table 1.** However, we did not separately analyze cancer risk in the excluded group. The difference of cancer risk in repeated screening and non-regular screening group is an interesting issue but it seems far from the topic of current research. However, we

investigated several gastrointestinal cancer risk by baseline HDL-C in 4.535 million and we found weak associations between baseline HDL-C and cancer risk (Supplementary Table 2). We added these results and Discussion. But if it is redundant, we can remove from article.

Supplementary Table 2. Gastrointestinal cancer risk by baseline HDL-C in 4.535 millions

	Gastric cancer		Liver cancer		Pancreatic cancer		Gallbladder-biliary cancer	
	aHR (95% CI)	P value	aHR (95% CI)	P value	aHR (95% CI)	P value	aHR (95% CI)	P value
Binary HDL-C (total)								
Low	1		1		1		1	
Normal	0.96 (0.94-0.98)	0.0003	0.97 (0.94-1.01)	0.137	0.95 (0.92-0.99)	0.025	1.01 (0.96-1.05)	0.734
Binary HDL-C (men)								
Low	1		1		1		1	
Normal	0.96 (0.94-0.99)	0.018	0.93 (0.89-0.97)	<0.001	0.90 (0.85-0.96)	0.002	0.96 (0.90-1.03)	0.238
Binary HDL-C(women)								
Low	1		1		1		1	
Normal	0.94 (0.90-0.97)	0.0006	1.03 (0.98-1.09)	0.254	0.98 (0.93-1.05)	0.623	1.02 (0.96-1.08)	0.468
5 HDL-C Category								
<30	1.03 (0.95-1.11)	0.495	1.69 (1.55-1.84)	<.0001	1.17 (1.02-1.355)	0.023	1.08 (0.94-1.25)	0.288
30-39	1.03 (0.99-1.06)	0.098	1.03 (0.98-1.08)	0.153	1.04 (0.97-1.10)	0.243	1.01 (0.95-1.07)	0.848
40-49	1		1		1		1	
50-59	0.97 (0.95-0.99)	0.019	1.02 (0.98-1.06)	0.250	1.00 (0.95-1.05)	0.998	1.01 (0.96-1.05)	0.785
≥60	0.97 (0.94-0.99)	0.006	1.14 (1.09-1.18)	<.0001	0.97 (0.93-1.02)	0.266	1.04 (0.99-1.08)	0.102

Adjusted for age, sex, body mass index, hypertension, diabetes, cerebrovascular disease, heart disease, smoking status, drinking status, lipid lowering drug, and physical activity. Liver factors (chronic hepatitis B, chronic hepatitis C, liver cirrhosis, and triglyceride) were additionally adjusted in liver cancer analysis. CI, confidence interval; HDL-C, high density lipoprotein cholesterol; HR, hazard ratio.

Low HDL-C refers to HDL-C < 40mg/dL in men and < 50mg/dL in women. Normal HDL-C means HDL-C ≥ 40mg/dL in men and ≥ 50mg/dL in women.

Results (p5)

Effect of baseline HDL-C on gastrointestinal cancer risk among 4.535 millions

In the adjusted analysis, normal HDL-C at baseline was associated with a reduced risk of gastric and pancreatic cancer comparing to low HDL-C in overall population men, and women. Gallbladder-biliary cancer risk had no association with baseline HDL-C. A normal HDL-C at baseline reduced the risk of men's liver cancer but had no association with women's liver cancer risk. Both extremely low (HDL-C < 30mg; aHR 1.69; 95% CI 1.55-1.84) and high HDL-C (HDL-C \geq 60mg; aHR 1.14; 95% CI 1.09-1.18) was associated with an increased risk of liver cancer (**Supplementary Table 2**).

Discussion (p9)

The link of HDL-C with liver cancer is interesting. Both extremely low (aHR 1.69) and high HDL-C at baseline (aHR 1.14) was associated with an increased risk of liver cancer. The persistent exposure to low HDL-C (HR 1.15), change from normal to low HDL-C (aHR 1.18), and change from low to normal HDL-C (HR 1.12) had a higher liver cancer risk compared to the persistently normal HDL-C group. **The finding that low-to-normal HDL-C level also increased the risk of cancer was observed only in liver cancer and was unexpected. Therefore, we further categorized HDL-C change group (Fig 4). A low-to-increase HDL-C change group (low HDL-C at baseline and Δ HDL-C over than 15 mg/dL) increased liver cancer risk, but low-to-normal HDL-C change group (low HDL-C at baseline and normal range of HDL-C at follow-up and Δ HDL-C less than 15 mg/dL) did not increased liver cancer risk. Furthermore, HDL-C level increase by > 15 mg/dL among the baseline normal HDL-C group (HR, 1.33) as well as the normal to low HDL-C group (HR, 1.22), increased liver cancer risk compared to the persistently normal group. Our findings suggest that extremely high HDL-C levels may increase the risk of liver cancer and emphasize that maintaining modest normal range of HDL-C levels is important to reduce the risk of liver cancer.**

Method (p 26)

Initially, we investigated the risk of gastrointestinal cancer according to baseline HDL-C in all population (4.535 millions).

We provided the baseline characteristics of included and excluded persons in **Supplementary Table 1**. However, the excluded persons included persons who did not undergo national cancer screening and individuals who expired or developed any cancer within 1 years after baseline enrollment. **We clarified the Method and provided baseline characteristics of included and excluded persons in Supplementary Table 1 and we additionally stated briefly in Result.**

Methods (p25).

From 2009 to 2017, NGHE provided measurement of HDL-C to all Korean persons aged 40 years or older. NHISS provided raw data after excluding persons with gastrointestinal cancer from 2004 to 2007. Subjects who both underwent NGHE (4.983 million) and cancer screening (4.75 million) in 2009 and NGHE in 2013 were included in this study. Among subjects who underwent both NGHE and National Cancer Screening from January to December 2009 (4.614million), we excluded persons with any C-codes within 12 months from index month (the month of health examination) (n=65,791) and the persons who died within 1 year of enrollment (n=14,863) (**Fig. 1A**). Death data from the National Statistical Office were also provided by the National Health Insurance Service. Nearly complete sensing of new cancer is possible through extracting C code form NHISS. In order to receive a special exemption for cancer, the cancer code should be registered with diagnostic evidence of cancer such as pathologic results at hospital. If any cancer code is registered, the patients pay just 5% for almost medical services. Therefore, cancer (C) code extracted from NHISS is highly reliable. Individuals who did not undergo NGHE in 2013 (n=1,665,999) or absence of HDL-C values were excluded [**model III; n=2,865,902**]. And persons who had been diagnosed with cancer (n=68,029) up to 2013 were further excluded [**model I, main model; n=2,797,873**] (**Fig 1B**).

Results (p4)

The baseline characteristics of the included and excluded individuals was provided in **Supplementary Table 1**. Included persons appears to have slightly better life styles (smoking and physical activities) and less presence of chronic disease such as hypertension and diabetes.

Supplementary Table 1. Baseline characteristics of included and excluded person

Included person (n=2,797,873)	Excluded person [‡] (n=2,185,555)
---

Age, year, mean (SD)	54.0 (9.9)	55.7 (11.2)
Economic status, mean (SD)*	11.7 (5.9)	11.8 (5.9)
BMI, kg/m ² , mean (SD)	24.0 (3.3)	24.0 (3.2)
HDL-C, mg/dL, mean (SD)	56.5 (34.9)	56.7 (37.3)
Men, no (%)	1333015 (47.6)	951166 (43.5)
Hypertension, no (%)	684077 (24.5)	595250 (27.1)
Heart disease, no (%)	64642 (2.3)	79407 (3.6)
Cerebrovascular disease, no (%)	26695 (1.0)	38538 (1.8)
Diabetes mellitus, no (%)	212653 (7.6)	205779 (9.4)
Drinking status, no (%)		
None	1608531 (58.3)	1341012 (61.7)
1/week	783990 (28.4)	517855 (23.8)
2-3/week	245933 (8.9)	193623 (8.9)
4-5/week	72236 (2.6)	69469 (3.2)
≥6/week	48617 (1.8)	52043 (2.4)
Smoking status, no (%)		
Never	1845429 (66.5)	1471730 (67.4)
Past	445372 (16.0)	288936 (13.2)
Current	486258 (17.5)	424621 (19.4)
Moderate activity, no (%) [†]		
None	1560935 (56.7)	1348350 (62.1)
1-2 day/week	605071 (22.0)	404482 (18.6)
3-5 day/week	443988 (16.1)	302755 (14.0)
6-7 day/week	142149 (5.2)	114805 (5.3)
Lipid lowering drug, no (%)	89105 (3.2)	60908 (2.8)
Liver factors, no (%)		
Any liver disease	47297 (2.4)	35510 (2.0)
Chronic hepatitis B	20547 (1.1)	11813 (0.7)
Chronic hepatitis C	11167 (0.6)	4193 (0.2)

BMI, body mass index; HDL-C, high density lipoprotein; SD, standard deviation.

*Economic status is twentile (1-20). 1 is the lowest and 20 is the highest income.

[†]Moderate physical activity refers to “walking or exercising and feeling mild dyspnea for more than 30 min per day.”

[‡]Excluded person among all persons who underwent baseline NGHE. This group includes persons who did not undergo national cancer screening and individuals who expired or

developed any cancer within 1 years after baseline enrollment.

2. For analysis of continuous change in HDL, the baseline HDL-c should be accounted for; I recommend stratifying and reporting the results for change seen among individuals with (a) normal HDL-c at baseline (b) low HDL-c at baseline, and (c) high HDL-c at baseline. (Also mentioned in my comments on “Results above)

Answer> Thank you for the important comment.

As reviewer’s comment, the effect of HDL change on cancer risk among individuals with (a) normal HDL-c at baseline (b) low HDL-c at baseline, and (c) high HDL-c at baseline can provide more information. We should conduct further analysis. However, it takes 1 year to get the raw data set again from NHISS. We are sorry for this limitation. Instead of this, **we provide the categorization B [(baseline normal-to-low, normal, increase) and (baseline low-to-low, normal, increase)] (Fig 1C, revised Fig 4).**

We further categorized HDL-C change using ATP III and absolute HDL-C change [$\Delta\text{HDL-C} = (\text{HDL-C at follow-up}) - (\text{HDL-C baseline})$] (**Fig. 1C and Supplementary material p5**). Persistent normal group was further classified into persistent normal and normal-increase (baseline normal and increase ≥ 15 mg/dL at follow-up from baseline) groups. **Baseline low group was classified as persistent low, low-to- normal, and low-to-increase.**

Fig 4

3. In the following sentence physical activity is listed twice: “We performed a multivariate analysis adjusted for significant confounders, such as age, sex, BMI, smoking status, alcohol intake, physical activity, economic status, use of lipid lowering drugs, hypertension, diabetes, cerebrovascular disease, heart disease, and physical activity”

Answer> Thank you for the kind comment. We revised the Methods (p26).

We performed a multivariate analysis adjusted for significant confounders, such as age, sex, BMI, smoking status, alcohol intake, economic status, use of lipid lowering drugs, hypertension, diabetes, cerebrovascular disease, heart disease, and physical activity.

4. Please list the endpoints/censorship variables in the section “Statistical Analysis”

Answer> Thank you for the kind comment. We clarified the Methods (p 26).

We calculated person-years from the date of baseline national general health examination to the following censoring events, whichever occurred first: occurrence of gastrointestinal cancer, death, or end of the study (December 31, 2017).

REVIEWERS' COMMENTS

Reviewer #1 (Remarks to the Author):

The authors tried to respond to the reviewer comments going in great length in their rebuttal letter and revised manuscript. I fully appreciate their efforts, but the majority (if not all) of the suggested extra analyses by the reviewers were not performed by the authors. Several reasons were listed and some other analyses were performed, but the main reason is the unavailability of the raw data, as their use has expired. The reviewer concerns and suggested extra analyses could have been easily addressed if the data were available. I still consider the mutual adjustment for other lipoproteins very important for this work, as they are correlated. The authors stressed HDL cholesterol as the important lipid for GI cancer risk, and they cite some papers in their reply that agree with this finding, but there are other published studies stressing that other lipids have clearer associations with the investigated cancers and that associations change after mutual adjustment. A simple mutual adjustment could have clarified this point in this dataset.

Reviewer #2 (Remarks to the Author):

Thank you to the authors for addressing all of my comments and taking the time to thoughtfully edit the manuscript. I think the revised paper is improved and the methods questions/limitations have been clarified.

RESPONSE TO REVIEWERS' COMMENTS

Thank you so much for giving the opportunity of the resubmission. Authors appreciate the critical and thoughtful review comments. We meticulously revised the manuscripts according to reviewers' comments and included point-by-point responses to each comment, stating clearly how the text has been changed. We indicated precisely where changes have been made in the manuscript, by changing the font to **blue color**. Red color is first revision.

We submitted this topic previously and we underwent major revision and changed manuscripts markedly. However, some additional analysis (exclusion period, consider LDL and TG as covariate, sex-smoking combination) commented by reviewers are unavailable because the raw data (2009 National Health Examination) were expired during waiting for review. We used available new raw data set from NHISS (this new data was claimed in 2022 and we can use this new data from October 15, 2023) and we conducted all analysis again using 2010 National Health Examination. As editor's recommendation, we newly submit the re-analyzed results. Additionally we added subgroup analysis by age group. The trend of association was similar to that of 2009 data set.

Reviewer #1 (Remarks to the Author):

The authors tried to respond to the reviewer comments going in great length in their rebuttal letter and revised manuscript. I fully appreciate their efforts, but the majority (if not all) of the suggested extra analyses by the reviewers were not performed by the authors. Several reasons were listed and some other analyses were performed, but the main reason is the unavailability of the raw data, as their use has expired. The reviewer concerns and suggested extra analyses

could have been easily addressed if the data were available. **I still consider the mutual adjustment for other lipoproteins very important for this work, as they are correlated.**

The authors stressed HDL cholesterol as the important lipid for GI cancer risk, and they cite some papers in their reply that agree with this finding, but there are other published studies stressing that other lipids have clearer associations with the investigated cancers and that associations change after mutual adjustment. **A simple mutual adjustment could have clarified this point in this dataset.**

Author reply) *Thank you for this important comments. Some additional analysis commented by reviewers are unavailable because the raw data (2009 National Health Examination) was expired during waiting for review. We can use new raw data set from NHISS (this new data was claimed in 2022 and we can use this new data from October 2023). We conducted all analysis again using 2010 National Health Examination (2010 baseline and follow-up to 2021). The trend of association after exclusion of any cancer up to 2014 (previous model) was similar to that of 2009 data set. Additional analysis after exclusion any cancer within 1 year or 2 year exclusion from baseline HDL-C measurement showed more significant association. Additionally we added subgroup analysis by age group.*

Blue color is points revised in second revision. Red color is part revised in first revision.

1) Issue for exclusion period

Reviewers commented that exclusion of any cancer up to 2nd measurement of HDL-C is over exclusion. Therefore, we conducted several models in the aspect of exposure period. We excluded any cancer with 1year from baseline (2010) (Main model), excluded any cancer with

2 year from baseline (model II), and excluded any cancer up to 2nd measurement of HDL-C (model III, main model in a first submission).

Analysis by exposure period

- *Main model (Table 2)*
- *Model II (Supplementary Table 3)*
- *Model III (Supplementary Table 4)*

Results (p6)

Sensitivity analysis

Our main results were robust to a range of sensitivity analyses for the impact of the exposure period in most cancers (**Table 2, Supplementary Table 3 and Supplementary Table 4**). The estimated effect sizes and CIs were in model II (**Supplementary Table 3**) similar to those in the model I, and the two graphs nearly overlapped in gastric, colorectal, liver, pancreatic, and gallbladder cancer risks (**Fig 2A and 2B**), wherein HDL-C change had no association with biliary cancer risk in model II. In adjusted analysis after further exclusion of cancer up to 2nd measurement of HDL-C [model III], the effect of HDL-C change on gastric, colorectal, and liver cancer risks was similar with that in model I, whereas HDL-C change had no significant effect on the risk of pancreatic, gallbladder, and biliary cancer (**Supplementary Table 4 and Fig 2A and 2C**).

Figure 2

Discussion (p 11)

Our main results were robust to a range of sensitivity analyses for the impact of exposure duration on most gastrointestinal cancers. The effects of HDL-C change on gastrointestinal cancers except biliary cancer were similar in model II (excluding any cancer within 2 years from baseline). The effects of HDL-C change on gastric, colorectal, and liver cancer was still significant but the effect on pancreatic, gallbladder, and biliary cancer risk was not significant in model III (excluding any cancer up to 2nd measurement of HDL-C).

2) Mutual adjustment for LDL and TG as covariate

We conducted all analysis again. We conducted all analysis additionally adjusted for LDL and TG. The impact of HDL change on most gastrointestinal cancer in the adjusted analysis that did not consider LDL and TG was similar to the results in the adjusted analysis that included LDL-C and TG. However, the effect of HDL-C change on biliary tract cancer risk was

significant in adjusted analyzes including LDL-C and TG, but not significant in adjusted analyzes excluding LDL-C and TG.

We showed the original analysis and additional analysis considering LDL and TG

Table 2. Gastrointestinal cancer risk by HDL-C change

HDL-C change	Number of case	Unadjusted HR		Adjusted HR*		Adjusted HR**	
		HR (95% CI)	P-value	HR (95% CI)	P-value	HR (95% CI)	P-value
Gastric cancer							
Low → Low	4,764	0.92 (0.89, 0.95)	<0.01	1.06 (1.03, 1.10)	<0.01	1.05 (1.02, 1.09)	<0.01
Low → Normal	4,061	0.94 (0.91, 0.97)	<0.01	1.04 (1.00, 1.07)	0.04	1.03 (0.99, 1.07)	0.11
Normal → Low	5,178	1.03 (1.00, 1.06)	0.09	1.05 (1.02, 1.09)	<0.01	1.05 (1.02, 1.08)	<0.01
Normal → Normal	26,693	1		1		1	
Colorectal cancer							
Low → Low	4,639	1.06 (1.03, 1.09)	<0.01	1.04 (1.00, 1.07)	0.03	1.02 (0.98, 1.05)	0.37
Low → Normal	3,580	0.98 (0.95, 1.02)	0.28	0.98 (0.95, 1.02)	0.37	0.97 (0.93, 1.00)	0.08
Normal → Low	4,919	1.15 (1.12, 1.19)	<0.01	1.09 (1.06, 1.13)	<0.01	1.08 (1.05, 1.12)	<0.01
Normal → Normal	22,569	1		1		1	
Liver cancer[†]							
Low → Low	2591	0.99 (0.95, 1.03)	0.52	1.07 (1.02, 1.12)	<0.01	1.17 (1.12, 1.23)	<0.01
Low → Normal	2029	0.93 (0.88, 0.97)	<0.01	0.99 (0.94, 1.04)	0.63	1.07 (1.02, 1.12)	0.01
Normal → Low	3138	1.22 (1.18, 1.27)	<0.01	1.21 (1.16, 1.27)	<0.01	1.26 (1.21, 1.32)	<0.01
Normal → Normal	13,551	1		1		1	
Pancreatic cancer							
Low → Low	1646	1.22 (1.16, 1.29)	<0.01	1.05 (0.99, 1.11)	0.12	1.05 (0.99, 1.12)	0.08
Low → Normal	1275	1.13 (1.07, 1.20)	<0.01	1.04 (0.97, 1.10)	0.25	1.04 (0.98, 1.11)	0.21
Normal → Low	1660	1.26 (1.20, 1.33)	<0.001	1.11 (1.05, 1.17)	<0.01	1.11 (1.05, 1.17)	<0.01
Normal → Normal	6,951	1		1		1	

Gallbladder cancer							
Low → Low	587	1.19 (1.09, 1.30)	<0.001	0.98 (0.89, 1.07)	0.61	0.97 (0.88, 1.07)	0.49
Low → Normal	465	1.13 (1.02, 1.25)	0.02	1 (0.91, 1.11)	0.95	1.00 (0.9, 1.11)	0.95
Normal → Low	634	1.32 (1.21, 1.44)	<0.001	1.13 (1.03, 1.23)	0.001	1.12 (1.02, 1.23)	0.01
Normal → Normal	2539	1		1		1	
Biliary cancer							
Low → Low	892	1.02 (0.95, 1.10)	0.54	0.94 (0.87, 1.02)	0.12	1.02 (1.00, 1.05)	0.03
Low → Normal	703	0.97 (0.89, 1.05)	0.38	0.94 (0.87, 1.02)	0.16	0.97 (0.95, 1.00)	0.02
Normal → Low	961	1.13 (1.05, 1.21)	<0.01	1.03 (0.96, 1.11)	0.44	1.11 (1.08, 1.13)	<0.01
Normal → Normal	4,495	1		1		1	

*Adjusted for age, sex, economic status, body mass index, hypertension, diabetes, cerebrovascular disease, heart disease, smoking status, drinking status, physical activity, and use of lipid lowering drug.

†Adjusted for age, sex, economic status, body mass index, hypertension, diabetes, cerebrovascular disease, heart disease, smoking status, drinking status, physical activity, use of lipid lowering drug, liver factors (chronic liver disease, chronic hepatitis B, chronic hepatitis C, and liver cirrhosis) and triglyceride.

**Adjusted for age, sex, economic status, body mass index, hypertension, diabetes, cerebrovascular disease, heart disease, smoking status, drinking status, physical activity, LDL, TG, and use of lipid lowering drug.

Supplementary Table 3. Sensitivity analysis I (Model II)

Supplementary Table 4. . Sensitivity analysis II (Model III)

Supplementary Table 7. Gastrointestinal cancer risk by absolute HDL-C change

Results

Impact of HDL-C change on gastrointestinal cancer risk

In the adjusted analysis for potential variables except triglyceride and LDL-C, the risk of gastric, colorectal, and liver cancers was higher in the persistently low HDL-C and normal-to-low HDL-C groups compared to the persistently normal HDL-C group and the risk of pancreatic and gallbladder cancer was higher in normal-to-low HDL-C groups compared to the persistently normal HDL-C group. In the adjusted analysis for all potential variables including triglyceride and LDL-C, these patterns persisted in most GI cancers (Table 2 and Fig. 2A). The persistently low HDL-C and normal-to-low HDL-C groups had a higher risk of gastric and biliary cancers compared to the persistently normal HDL-C group. The persistently low (adjusted HR [aHR] 1.17), normal-to-low (aHR 1.26), and low-to-normal (aHR 1.07) HDL-C groups had a higher risk of liver cancer compared to the persistently normal HDL-C group. Normal-to-low HDL-C groups had a higher risk of colorectal (aHR 1.08), pancreatic (aHR 1.11), and gallbladder (aHR 1.12) cancer compared to the persistently normal HDL-C group.

We showed analysis additionally considering LDL and TG

Extended Data Table 1. Sub group analysis by sex (adjusted analysis)

	HDL-C change	Men		Women	
		HR (95% CI) *	P value	HR (95% CI) *	P value
Gastric cancer	Low → Low	1.04 (0.99, 1.09)	0.11	1.11 (1.06, 1.16)	<0.01
	Low → Normal	1.02 (0.98, 1.07)	0.38	1.07 (1.02, 1.13)	0.01
	Normal → Low	1.05 (1.01, 1.09)	0.02	1.07 (1.02, 1.12)	0.01
	Normal → Normal	1		1	
Liver cancer[†]	Low → Low	1.06 (0.99, 1.13)	0.1	1.08 (1.01, 1.15)	0.03
	Low → Normal	1.00 (0.94, 1.07)	0.92	0.97 (0.89, 1.05)	0.40

	Normal → Low	1.24 (1.18, 1.30)	<0.01	1.18 (1.09, 1.26)	<0.01
	Normal → Normal	1		1	
Colorectal cancer	Low → Low	1.04 (0.98, 1.1)	0.22	1.06 (1.01, 1.10)	0.01
	Low → Normal	0.97 (0.92, 1.02)	0.26	1.00 (0.95, 1.05)	0.95
	Normal → Low	1.1 (1.05, 1.15)	<0.01	1.09 (1.04, 1.14)	<0.01
	Normal → Normal	1		1	
Pancreatic cancer	Low → Low	0.99 (0.89, 1.10)	0.77	1.08 (1.00, 1.15)	0.04
	Low → Normal	1.12 (1.02, 1.23)	0.02	1.00 (0.92, 1.08)	0.91
	Normal → Low	1.10 (1.01, 1.19)	0.03	1.12 (1.03, 1.20)	<0.01
	Normal → Normal	1		1	
Gallbladder cancer	Low → Low	0.92 (0.82, 1.05)	0.22	1.04 (0.93, 1.16)	0.51
	Low → Normal	0.97 (0.86, 1.09)	0.55	1.00 (0.88, 1.15)	0.98
	Normal → Low	0.98 (0.89, 1.08)	0.68	1.04 (0.92, 1.18)	0.53
	Normal → Normal	1		1	
Biliary cancer	Low → Low	0.92 (0.82, 1.05)	0.22	0.96 (0.87, 1.06)	0.45
	Low → Normal	0.97 (0.86, 1.09)	0.55	0.93 (0.83, 1.05)	0.24
	Normal → Low	0.98 (0.89, 1.08)	0.68	1.09 (0.98, 1.21)	0.11
	Normal → Normal	1		1	

*Adjusted for age, economic status, body mass index, hypertension, diabetes, cerebrovascular disease, heart disease, smoking status, drinking status, physical activity, **LDL**, **TG**, and use of lipid lowering drug.

†Adjusted for age, sex, economic status, body mass index, hypertension, diabetes, cerebrovascular disease, heart disease, smoking status, drinking status, physical activity, use of lipid lowering drug, **LDL**, **TG**, and liver factors (chronic liver disease, chronic hepatitis B, chronic hepatitis C, and liver cirrhosis).

Age, body mass index, and economic status were continuous variables.

CI, confidence interval; HDL-C, high density lipoprotein cholesterol; HR, hazard ratio. Low HDL-C refers to <40mg/dL in men and <50mg/dL in women. Normal HDL-C means ≥40mg/dL in men and ≥50mg/dL in women.

Extended Data Table 2. Effect of HDL-C change by smoking status (adjusted analysis)

Extended Data Table 3. Subgroup analysis by combination of age and smoking status

Extended Data Table 4. Effect of HDL-C change by age (adjusted analysis)

Supplementary Table 6. Gastrointestinal cancer risk by further categorization of HDL-C change

Discussion (p 14)

The impact of HDL-C change on most gastrointestinal cancer in the adjusted analysis that did not consider LDL and TG was similar to the results in the adjusted analysis that included LDL-C and TG. However, the effect of HDL-C change on biliary tract cancer risk was significant in adjusted analyzes including LDL-C and TG, but not significant in adjusted analyzes excluding LDL-C and TG. These results suggested that HDL-C change affect gastrointestinal cancer risk regardless of other lipid level.

3) Subgroup analysis by age group.

Results (p8)

Subgroup analysis by age groups

Interaction analysis showed a significant interaction between age groups and HDL change in the risk of gastrointestinal cancers. In adjusted analysis, the impact of HDL-C change on several gastrointestinal cancers showed an age discrepancy (**Extended Data Table 4 and Extended Data Fig 1**). The effect of HDL-C change on the risk of gastric and gallbladder cancer was significant only in age ≥ 60 years but not significant in age < 60 years. The effect of HDL-C change on the risk of liver and colorectal cancer risk was significant in both age groups. The effect of HDL-C change on the risk of pancreatic and biliary cancer was significant only in age < 60 years, but not significant in age ≥ 60 years.

Extended Data Table 4. Effect of HDL-C change by age (adjusted analysis)

	HDL-C change	<60yr		≥60yr	
		HR (95% CI)*	P value	HR (95% CI)*	P value
Gastric cancer	Low → Low	1.04 (0.98, 1.10)	0.17	1.07 (1.03, 1.12)	0.001
	Low → Normal	1.03 (0.97, 1.09)	0.31	1.04 (0.99, 1.08)	0.13
	Normal → Low	1.03 (0.98, 1.08)	0.29	1.06 (1.02, 1.11)	0.002
	Normal → Normal	1		1.00	
Liver cancer†	Low → Low	1.21 (1.12, 1.30)	<.0001	1.14 (1.07, 1.21)	<.0001
	Low → Normal	1.02 (0.95, 1.11)	0.57	1.09 (1.02, 1.17)	0.01
	Normal → Low	1.32 (1.23, 1.41)	<.0001	1.22 (1.16, 1.29)	<.0001
	Normal → Normal	1		1.00	
Colorectal cancer	Low → Low	1.06 (.001, 1.12)	0.04	1.00 (0.96, 1.05)	0.96
	Low → Normal	0.95 (0.90, 1.01)	0.10	0.98 (0.94, 1.03)	0.43
	Normal → Low	1.06 (1.01, 1.12)	0.02	1.09 (1.05, 1.14)	<.0001
	Normal → Normal	1		1.00	
Pancreatic cancer	Low → Low	1.11 (1.01, 1.23)	0.04	1.03 (0.96, 1.11)	0.47
	Low → Normal	1.11 (1.00, 1.23)	0.05	1.01 (0.93, 1.09)	0.87
	Normal → Low	1.21 (1.10, 1.32)	<.0001	1.05 (0.98, 1.13)	0.16
	Normal → Normal	1		1.00	
Gallbladder cancer	Low → Low	1.14 (0.95, 1.36)	0.15	0.91 (0.81, 1.03)	0.12
	Low → Normal	1.04 (0.86, 1.27)	0.67	0.98 (0.87, 1.11)	0.76
	Normal → Low	1.12 (0.94, 1.33)	0.21	1.12 (1.01, 1.25)	0.03
	Normal → Normal	1		1.00	
Biliary cancer	Low → Low	0.95 (0.82, 1.11)	0.53	0.95 (0.87, 1.04)	0.27
	Low → Normal	0.95 (0.81, 1.11)	0.53	0.95 (0.86, 1.05)	0.31
	Normal → Low	1.19 (1.04, 1.36)	0.01	0.98 (0.9, 1.07)	0.60
	Normal → Normal	1		1.00	

*Adjusted for sex, economic status, body mass index, hypertension, diabetes, cerebrovascular disease, heart disease, drinking status, smoking status, physical activity, **LDL**, **TG**, and use of lipid lowering drug.

†Adjusted for age, sex, economic status, body mass index, hypertension, diabetes, cerebrovascular disease, heart disease, drinking status, smoking status, physical activity, use of lipid lowering drug, **LDL**, **TG**, and liver factors (chronic liver disease, chronic hepatitis B, chronic hepatitis C, and liver cirrhosis).

Age, body mass index, and economic status were continuous variables.

CI, confidence interval; HDL-C, high density lipoprotein cholesterol; HR, hazard ratio.

Low HDL-C refers to <40mg/dL in men and <50mg/dL in women. Normal HDL-C means ≥40mg/dL in men and ≥50mg/dL in women.

Extended Data Fig 1. Subgroup analysis by age group

Extended fig 1

Discussion (11-12)

Interestingly, subgroup analysis by age group showed a discrepant effect on individual gastrointestinal cancers. The effect of HDL-C change on the risk of gastric and gallbladder cancer was significant only in age ≥ 60 years but not significant in age < 60 years. The effect of HDL-C change on the risk of pancreatic and biliary cancer was significant only in age < 60 years, but not significant in age ≥ 60 years. The effect of HDL-C change on the risk of liver and colorectal cancer risk was significant in both age groups. The mechanism of site specific age discrepancy in the association between HDL-C change and digestive cancer risk is unknown. A different peak onset ages and site-specific biologic response to low HDL-C may partially contribute to this site specific age discrepancy in the association between HDL-C changes and gastrointestinal cancer risk.

Reviewer #2 (Remarks to the Author):

Thank you to the authors for addressing all of my comments and taking the time to

thoughtfully edit the manuscript. I think the revised paper is improved and the methods questions/limitations have been clarified.

Author reply) *Thank you for the generous comments. However, some additional analysis commented by reviewers are unavailable because the raw data (2009 National Health Examination) was expired during waiting for review. We can use new raw data set from NHISS (this new data were claimed in 2022 and we can use this new data from October 2023). We conducted all analysis again using 2010 National Health Examination (2010 baseline and follow-up to 2021). The trend of association after exclusion of any cancer up to 2014 (previous model) was similar to that of 2009 data set. Additional analysis after exclusion any cancer within 1 year or 2 year exclusion from baseline HDL-C measurement showed more significant association. We additionally conducted subgroup analysis by sex-smoking combination and adjusted analysis in the effect of absolute HDL-C change on cancer risk.*

Blue color is points revised in second revision. Red color is part revised in first revision.

1) sex-smoking combination analysis

We conducted subgroup analysis by combination of sex and smoking.

Extended Data Table 3. Subgroup analysis by combination of age and smoking status

	HDL-C change	men-smoker		men-never smoker	
		HR (95% CI) *	P value	HR (95% CI) *	P value
Gastric cancer	Low → Low	1.08 (1.02, 1.14)	0.014	0.97 (0.89, 1.06)	0.463
	Low → Normal	1.02 (0.97, 1.08)	0.409	1.01 (0.93, 1.09)	0.910
	Normal → Low	1.06 (1.01, 1.11)	0.015	1.06 (0.99, 1.14)	0.101

	Normal→ Normal	1		1	
Liver cancer[†]	Low → Low	1.23 (1.13, 1.33)	<.0001	1.07 (0.95, 1.21)	0.246
	Low → Normal	1.09 (1.01, 1.18)	0.034	1.09 (0.98, 1.22)	0.119
	Normal → Low	1.37 (1.28, 1.45)	<.0001	1.21 (1.1, 1.33)	<.0001
	Normal→ Normal	1		1	
Colorectal cancer	Low → Low	1.03 (0.96, 1.1)	0.409	1.01 (0.92, 1.11)	0.850
	Low → Normal	0.95 (0.89, 1.02)	0.134	0.97 (0.88, 1.06)	0.473
	Normal → Low	1.11 (1.05, 1.17)	0.000	1.09 (1.01, 1.18)	0.026
	Normal→ Normal	1		1	
Pancreatic cancer	Low → Low	1.03 (0.9, 1.18)	0.664	1.01 (0.84, 1.21)	0.936
	Low → Normal	1.22 (1.09, 1.37)	0.001	1 (0.84, 1.19)	0.962
	Normal → Low	1.15 (1.03, 1.27)	0.010	1.09 (0.95, 1.26)	0.231
	Normal→ Normal	1		1	
Gallbladder cancer	Low → Low	0.81 (0.63, 1.03)	0.088	0.74 (0.53, 1.03)	0.078
	Low → Normal	1.04 (0.85, 1.28)	0.705	0.94 (0.71, 1.24)	0.655
	Normal → Low	1.32 (1.12, 1.56)	0.001	1.15 (0.92, 1.43)	0.216
	Normal→ Normal	1		1	
Biliary cancer	Low → Low	1.01 (0.86, 1.18)	0.944	0.86 (0.69, 1.07)	0.182
	Low → Normal	0.96 (0.82, 1.11)	0.574	1.03 (0.85, 1.25)	0.750
	Normal → Low	0.99 (0.87, 1.12)	0.856	1.03 (0.88, 1.21)	0.720
	Normal→ Normal	1		1	
		women-smoker		women-never smoker	
	HDL-C change	HR (95% CI) *	P value	HR (95% CI) *	P value

Gastric cancer	Low → Low	1.01 (0.79, 1.3)	0.926	1.11 (1.06, 1.17)	<.0001
	Low → Normal	1.15 (0.88, 1.5)	0.299	1.07 (1.01, 1.13)	0.023
	Normal → Low	1.19 (0.93, 1.52)	0.175	1.07 (1.01, 1.12)	0.015
	Normal → Normal	1		1	
Liver cancer[†]	Low → Low	1.6 (1.15, 2.23)	0.005	1.2 (1.12, 1.29)	<.0001
	Low → Normal	1.46 (1.01, 2.1)	0.045	1.04 (0.96, 1.13)	0.318
	Normal → Low	1.23 (0.85, 1.76)	0.271	1.23 (1.14, 1.32)	<.0001
	Normal → Normal	1		1	
Colorectal cancer	Low → Low	0.93 (0.74, 1.16)	0.497	1.05 (1, 1.1)	0.048
	Low → Normal	0.95 (0.74, 1.21)	0.679	0.99 (0.94, 1.04)	0.711
	Normal → Low	1.03 (0.83, 1.29)	0.782	1.09 (1.04, 1.14)	0.001
	Normal → Normal	1		1	
Pancreatic cancer	Low → Low	0.92 (0.64, 1.34)	0.677	1.08 (1.01, 1.17)	0.034
	Low → Normal	0.81 (0.52, 1.26)	0.349	1 (0.92, 1.09)	0.993
	Normal → Low	1.01 (0.69, 1.47)	0.972	1.12 (1.04, 1.21)	0.003
	Normal → Normal	1		1	
Gallbladder cancer	Low → Low	0.99 (0.53, 1.88)	0.986	1.04 (0.93, 1.18)	0.487
	Low → Normal	1.2 (0.61, 2.37)	0.604	0.99 (0.87, 1.14)	0.928
	Normal → Low	1.36 (0.73, 2.54)	0.340	1.03 (0.91, 1.17)	0.626
	Normal → Normal	1		1	

Biliary cancer	Low → Low	0.71 (0.39, 1.27)	0.243	0.99 (0.89, 1.09)	0.777
	Low → Normal	1.48 (0.89, 2.46)	0.135	0.92 (0.81, 1.04)	0.163
	Normal → Low	1.68 (1.06, 2.67)	0.027	1.07 (0.96, 1.19)	0.217
	Normal → Normal	1		1	

Results (P 8)

Subgroup analysis by combination of sex and smoking

We conducted subgroup analysis using a combination of sex and smoking (**Extended Data Table 3**). Persistent low, low-to-normal, and normal-to low HDL-C change slightly increased the gastric cancer risk in never smoking women and persistent low and normal-to low HDL-C change increased the gastric cancer risk in smoking men. These patterns were also observed in pancreatic cancer risk. In men, normal-to-low HDL-C group was associated with an increased risk of CRC in both never smokers and ever-smokers, whereas in women persistent low and normal-to low HDL-C change increased CRC risk in never smokers.

Persistent low, low-to-normal, and normal-to low HDL-C change markedly increased the liver cancer risk in smoking men. Persistent low and normal-to low HDL-C change increased the liver cancer risk in never smoking women and persistent low and low-to-normal HDL-C markedly increased the liver cancer risk in never smoking women.

Discussion (P 12)

In subgroup analysis by combination of sex and smoking status (men-smoker, men-never smoker, women-smoker, women-never smoker), the effect of HDL-C change on

women's gastric cancer risk was significant only in never smokers and the effect of HDL-C change on men's gastric cancer risk was significant only in smokers. These patterns were observed in pancreatic cancer. The hazardous effect of persistent low or normal-to-low HDL-C on liver cancer risk was observed in all 4 groups, but effect size was markedly high in smokers regardless of sex. The mechanism of this discrepant effect of HDL-C change according to sex and smoking status in several cancers is unknown. A plausible explanation is that the immunologic and inflammatory response to smoking further promotes a low HDL-C level-related cancer risk in current smokers. The related mechanisms need to be investigated in the future. Previous epidemiologic studies have reported sex-discrepant effects of BMI on cancer risk.²⁰ The interaction between BMI and smoking status on cancer risk has also been previously reported.^{20,21} **In a previous small study (259 cases of lung cancer in 14,547 members of atherosclerotic risk) using binary HDL-C, low HDL-C at baseline was associated with lung cancer only in past smokers (HR 1.77).²²**

2) Adjusted analysis in the effect of absolute HDL-C change on cancer risk

Results (8-9)

Effect of absolute HDL-C change on cancer risk

In unadjusted analysis, a decrement of HDL-C at follow-up increases the risk of gastric, colorectal, liver, pancreatic, gallbladder, and biliary cancers comparing to stable HDL-C group. In adjusted analysis, a decrement of HDL-C at follow-up (Δ HDL-C < -10 mg/dL) increases the risk of gastric, colorectal, liver, pancreatic, gallbladder, and biliary cancer (Supplementary Table 7 and Fig 5). Interestingly, marked increase of HDL-C (Δ HDL-C \geq 25 mg/dL) also associated with an increased risk of gastric (aHR 1.07; 95% CI 1.00-1.14) and liver cancer (aHR 1.41; 95% CI 1.3-1.53), whereas increase of HDL-C slightly reduced the

risk of colorectal cancer risk (aHR 0.94; 95% CI 0.90-0.98).

Discussion (p 13)

We also investigated the effect of absolute HDL-C change on cancer risk. A decrement of absolute HDL-C level (Δ HDL-C < -10 mg/dL) increased the risk of gastric, colorectal, pancreatic, gallbladder, and biliary cancers comparing to stable HDL-C group. An increment of HDL-C from baseline (Δ HDL-C \geq 25 mg/dL) was also associated with an increased risk of gastric (aHR 1.07) and liver cancer (aHR 1.41), whereas an increment of HDL-C (Δ HDL-C = 5-25 mg/dL) slightly reduced the risk of colorectal cancer (aHR 0.94). These results emphasize the importance of avoiding a decrement of HDL-C to prevent all digestive cancers and encourage an increment of HDL-C to prevent colorectal cancer.

Supplementary Table 7. Gastrointestinal cancer risk by absolute HDL-C change

	HDL-C change	Number of case	Unadjusted HR		Adjusted HR*		Adjusted HR**	
			HR (95% CI)	p-value	HR (95% CI)	p-value	HR (95% CI)	p-value
Gastric cancer	Δ HDL<-10	7,747	1.1 (1.07, 1.13)	<.0001	1.05 (1.02, 1.08)	0.003	1.05 (1.02, 1.08)	0.002
	Δ HDL: -10~ -5	5,573	1.03 (1, 1.06)	0.051	1.02 (1.05)	0.273	1.02 (0.99, 1.05)	0.236
	Δ HDL: -5~5	14,327	1		1		1	
	Δ HDL: 5-15	8,969	1 (0.97, 1.02)	0.745	1.03 (1, 1.05)	0.077	1.02 (1, 1.05)	0.093
	Δ HDL:15-25	2,944	0.98 (0.94, 1.02)	0.318	1.03 (0.99, 1.08)	0.130	1.03 (0.99, 1.07)	0.147
	Δ HDL \geq 25	1,136	1.04 (0.98, 1.1)	0.239	1.07 (1, 1.14)	0.036	1.07 (1, 1.14)	0.038
Colorectal cancer	Δ HDL<-10	7,143	1.14 (1.11, 1.18)	<.0001	1.07 (1.04, 1.1)	<.0001	1.07 (1.04, 1.11)	<.0001
	Δ HDL: -10~ -5	4,978	1.04 (1.01, 1.08)	0.013	1.03 (0.99, 1.06)	0.146	1.03 (0.99, 1.06)	0.106
	Δ HDL: -5~5	12,663	1		1		1	
	Δ HDL: 5-15	7,535	0.95 (0.92, 0.97)	0.000	0.96 (0.93, 0.99)	0.008	0.96 (0.93, 0.99)	0.006
	Δ HDL:15-25	2,447	0.92 (0.88, 0.97)	0.000	0.94 (0.9, 0.98)	0.004	0.94 (0.9, 0.98)	0.003

			0.96)							
	Δ HDL \geq 25	941	0.97 (1.04)	(0.91, 0.370	0.97 (1.04)	(0.91, 0.357	0.97 (0.9, 1.03)	0.322		
Liver cancer[†]	Δ HDL $<$ -10	4,829	1.39 (1.44)	(1.34, <.0001	1.32 (1.37)	(1.27, <.0001	1.28 (1.24, 1.34)	<.0001		
	Δ HDL: -10~ - 5	2,831	1.06 (1.11)	(1.02, 0.006	1.05 (1, 1.1)	0.051	1.04 (0.99, 1.09)	0.113		
	Δ HDL: -5~5	7,065	1		1		1			
	Δ HDL: 5-15	4,312	0.97 (1.01)	(0.93, 0.112	1.01 (1.05)	(0.97, 0.805	1.01 (0.97, 1.05)	0.601		
	Δ HDL:15-25	1,558	1.05 (1.11)	(0.99, 0.083	1.13 (1.07, 1.2)	<.0001	1.14 (1.07, 1.21)	<.0001		
	Δ HDL \geq 25	714	1.32 (1.42)	(1.22, <.0001	1.39 (1.51)	(1.28, <.0001	1.41 (1.3, 1.53)	<.0001		
Pancreatic cancer	Δ HDL $<$ -10	2,298	1.16 (1.1, 1.22)	<.0001	1.07 (1.13)	(1.02, 0.009	1.07 (1.02, 1.13)	0.009		
	Δ HDL: -10~ - 5	1,576	1.04 (0.98, 1.1)	0.171	1.01 (1.07)	(0.95, 0.741	1.01 (0.95, 1.07)	0.750		
	Δ HDL: -5~5	4,015	1		1		1			
	Δ HDL: 5-15	2,470	0.98 (1.03)	(0.93, 0.363	0.98 (1.04)	(0.93, 0.523	0.98 (0.94, 1.04)	0.530		
	Δ HDL:15-25	850	1.01 (1.08)	(0.94, 0.853	1 (0.93, 1.08)	0.967	1 (0.93, 1.08)	0.973		
	Δ HDL \geq 25	323	1.05 (1.18)	(0.94, 0.405	1.05 (1.18)	(0.94, 0.405	1.05 (0.94, 1.18)	0.403		
Gallbladder cancer	Δ HDL $<$ -10	862	1.21 (1.31)	(1.11, <.0001	1.1 (1.01, 1.2)	0.028	1.1 (1.01, 1.2)	0.027		
	Δ HDL: -10~ - 5	588	1.07 (1.18)	(0.98, 0.146	1.04 (1.15)	(0.94, 0.416	1.04 (0.95, 1.15)	0.404		
	Δ HDL: -5~5	1,453	1		1		1			
	Δ HDL: 5-15	889	0.97 (1.06)	(0.89, 0.495	0.99 (1.08)	(0.91, 0.767	0.99 (0.91, 1.07)	0.748		
	Δ HDL:15-25	305	1 (0.88, 1.13)	0.972	0.98 (1.11)	(0.86, 0.733	0.98 (0.86, 1.11)	0.718		
	Δ HDL \geq 25	128	1.15 (1.38)	(0.96, 0.135	1.13 (1.36)	(0.94, 0.199	1.13 (0.94, 1.36)	0.199		
Biliary cancer	Δ HDL $<$ -10	1,422	1.18 (1.1, 1.26)	<.0001	1.1 (1.02, 1.17)	0.008	1.09 (1.02, 1.17)	0.010		
	Δ HDL: -10~ - 5	907	0.98 (1.06)	(0.91, 0.632	0.97 (1.05)	(0.89, 0.391	0.97 (0.89, 1.04)	0.372		
	Δ HDL: -5~5	2,452	1		1		1			
	Δ HDL: 5-15	1,588	1.03 (0.97, 1.1)	0.385	1.06 (1.13)	(0.99, 0.080	1.06 (0.99, 1.13)	0.076		
	Δ HDL:15-25	485	0.94 (1.04)	(0.85, 0.220	0.97 (1.07)	(0.88, 0.509	0.97 (0.88, 1.07)	0.520		
	Δ HDL \geq 25	197	1.05 (1.21)	(0.91, 0.528	1.08 (1.26)	(0.94, 0.283	1.09 (0.94, 1.26)	0.270		

Fig 5

REVIEWERS' COMMENTS

Reviewer #1 (Remarks to the Author):

The authors have appropriately addressed all comments and concerns. Thank you!